# Dynamic Orthogonal Continual Fine-tuning for Mitigating Catastrophic Forgettings

## Abstract

Catastrophic forgetting remains a critical challenge in continual learning for large language models (LLMs), where models struggle to retain performance on historical tasks when fine-tuning on new sequential data without access to past datasets. In this paper, we first reveal that the drift of functional directions during the fine-tuning process is a key reason why existing regularization-based methods fail in long-term LLM continual learning. To address this, we propose Dynamic Orthogonal Continual (DOC) fine-tuning, a novel approach that tracks the drift of these functional directions and dynamically updates them during the fine-tuning process. Furthermore, by adjusting the gradients of new task parameters to be orthogonal to the tracked historical function directions, our method mitigates interference between new and old tasks. Extensive experiments on various LLM continual learning benchmarks demonstrate that this approach outperforms prior methods, effectively reducing catastrophic forgetting and providing a robust tool for continuous LLM fine-tuning.

## 1 Introduction

Recently, Large Language Models (LLMs) have achieved significant milestones in various tasks based on their extensive capacity and knowledge. In particular, fine-tuning LLMs with task-specific data has emerged as a popular learning paradigm in their diverse applications. In this context, LLM Continual Learning (Wu et al., 2024b), which fine-tunes LLMs with evolving tasks and data, has become a crucial technique for updating their knowledge to keep pace with new environments and goals. However, a critical challenge of continual learning is catastrophic forgetting (Wu et al., 2022), where the model forgets the knowledge it acquired from previous tasks after receiving new updates.

Existing continual learning approaches for LLMs can be categorized into the following types (Wu et al., 2024b; Zheng et al., 2025): Rehearsal-based (de Masson D'Autume et al., 2019; Mok et al., 2023; Huang et al., 2021), Architecture-based Jang et al. (2023); Peng et al. (2024); Wu et al. (2024a), Prompt-based (Wang et al., 2022; Qin & Joty, 2022; Razdaibiedina et al., 2023), and Regularization-based approaches (Farajtabar et al., 2020; Wang et al., 2023a; Li & Hoiem, 2017; Kirkpatrick et al., 2017; Zenke et al., 2017). While the first three approaches may suffer from significant computational or memory overhead issues (*e.g.*, training additional modules or storing historical data), regularization-based continual learning for LLMs does not suffer from these issues and has been acknowledged as an efficient approach (Wu et al., 2024b). More formally, we denote that all existing regularization-based methods abide by the following outline:

- **Step (1)** Record the functional directions, mainly the *gradient direction* of the model parameter, on historical tasks (Olah et al., 2020; 2018; Saxena & Cunningham, 2019);
- **Step (2)** Regularize new updates based on these historical functional directions.

For instance, Elastic Weight Consolidation (EWC) methods (Kirkpatrick et al., 2017; Zenke et al., 2017) and orthogonal optimization methods, including Orthogonal Gradient Descent (OGD) (Farajtabar et al., 2020) and Orthogonal Subspace Learning (O-LoRA) (Wang et al., 2023a), employ historical gradient directions and vectors in LoRA matrices of the model for regularization.

However, current regularization-based continual learning still faces the catastrophic forgetting problem, leaving a gap for its practical deployment. In this paper, we aim to mitigate this problem

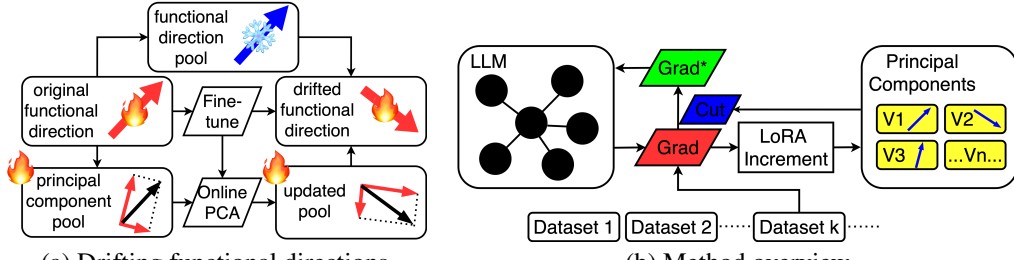

(a) Drifting functional directions        (b) Method overview

Figure 1: An introduction to our work. Figure (a) demonstrates the motivation of our method. Prior methods record the functional directions in a *fixed* pool and try to regularize future updates with it, which is shown on the upper half of Figure (a). Our method (the lower half) *updates* these directions with Online PCA for better regularization. Figure (b) presents an overview of our method. In a sequence of incoming datasets, we compute gradients and LoRA increment (described in Equation (5)) to update a set of principal components that represent drifting functional directions (described in line 4 of Algorithm 1). We cut them off from current gradients to avoid forgetting historical functions (described in Equation (13)).

by identifying a key problem in the regularizers. Specifically, we find that the drift of functional directions (Black et al., 2022) during continuous fine-tuning poses a significant issue for their regularizations. While functional directions may be valid within a local neighborhood around a static point in the parameter space, continuous fine-tuning can break this locality when moving the model weights towards other spaces, thus destroying the functionality of these directions, as shown in Figure 1(a). This observation is detailed in Section 3.1. In the settings of regularization-based methods, the difficulty lies in the lack of access to historical data, which makes it challenging to update their functional directions in the current parameter space.

Based on the observation above, we propose our method that tracks the drifting functional directions of historical tasks with the latest task data. Since LLMs primarily fine-tune within a low-rank subspace (Aghajanyan et al., 2021), all tasks share most of the functional directions in this subspace with different linear combinations. Thus, we employ a modified Online Principal Component Analysis (Cardot & Degras, 2018) to extract these directions from their combinations to capture and track the evolving functional directions. Leveraging these up-to-date functional directions, we cut gradients of new task parameters to be orthogonal to the tracked historical function directions, following prior orthogonal methods including OGD (Farajtabar et al., 2020) and O-LoRA (Wang et al., 2023a), which mitigates the interference between new and old tasks. However, a key difference between our method and other orthogonal methods is that we dynamically *update* the functional directions rather than regularizing on *fixed* ones. Tracking these functional directions, which prior works often overlook, is crucial for preserving functions that lie in drifting directions. A brief overview of our method is in Figure 1(b).

Extensive experiments verify the drift of functional directions and demonstrate the effectiveness of our method in tracking them, offering a substantiated motivation for our method. Furthermore, experiments on various LLM continual learning benchmarks demonstrate that our approach significantly mitigates the catastrophic forgetting issues in online streaming data scenarios, and outperforms prior methods, *e.g.*, we respectively achieve an accuracy of 77.7 and 73.4 in standard CL benchmark (Zhang et al., 2015) and long chains of tasks for LLaMA-7B (Touvron et al., 2023), compared to 76.5 and 71.9 of O-LoRA (Wang et al., 2023a), the previous state-of-the-art regularization-based method. In summary, our contributions are as follows:

- We reveal the drift of function directions in the fine-tuning process, which explains why regularization-based approaches fail in long-term LLM continual learning.

- Based on this discovery, we propose the Dynamic Orthogonal Continual Fine-tuning (DOC) method that tracks the drift of functional directions to mitigate catastrophic forgetting issues.

- Extensive experiments show that DOC outperforms prior methods in various LLM continual learning benchmarks, contributing an effective tool in continuous LLM fine-tuning.

## 2 PRELIMINARIES

### 2.1 CONTINUAL LEARNING SETUP

Continual learning for LLMs (Wu et al., 2024b; Zheng et al., 2025) is crucial for updating their knowledge and keeping pace with new goals. In a continual learning scenario, a pre-trained LLM is fine-tuned on an online stream of tasks with their task-specific data. Due to factors like storage costs and privacy protection, historical data cannot be accessed when fine-tuning on the latest one.

**Definition of continual learning**. Given a LLM $F_\theta$ with parameters $\theta$, a sequence of labeled datasets $\{D_1, D_2, ..., D_N\}$, where $D_t = \{(x_t^i, y_t^i)\}_{i=1}^{n_t}$ $(t = 1, ..., N)$. Then $F_\theta$ is sequentially fine-tuned on $D_1, D_2, ..., D_N$. When fine-tuning on $D_T$, historical datasets, *i.e.* $\{D_1, D_2, ..., D_{t-1}\}$, cannot be accessed. The target is an $F_\theta$ that behaves well on all datasets:

$$\arg\min_\theta \sum_{t=1}^{N} \sum_{i=1}^{n_i} L_t(F_\theta(x_t^i), y_t^i), \tag{1}$$

where $L_t$ is the task-specific loss function of the $t$-th task. **Note that for concision, we substitute $L$ for $L_t$ when fine-tuning on $D_t$ in the following statements**.

### 2.2 LOW-RANK ADAPTATION (LoRA)

Our method is developed for continual fine-tuning using LoRA, and maintaining one set of LoRA modules in this process. When fine-tuning LLMs for specific tasks, there exists a low intrinsic dimension for the parameter update of the model (Aghajanyan et al., 2021). For a weight matrix $W_{m \times n}$ of a pre-trained LLM, LoRA (Hu et al., 2022) employs low-rank matrixes $B_{m \times r}$ and $A_{r \times n}$ $(r \ll min(m, n))$ to constrain its update by representing it with a low-rank decomposition:

$$W^* = W + BA, \tag{2}$$

where $W^*$ is the new parameter after fine-tuning. As a result, the propagation process is modified:

$$W^*x = (W + BA)x = Wx + BAx, \tag{3}$$

where $x$ is the input to the module with parameter $W$.

## 3 MOTIVATION AND THE PROPOSED METHOD

In this section, we first reveal that the drift of functional directions is the key issue for existing regularization-based methods in 3.1, then propose a method to track drifting functional directions and validate the effectiveness of our tracking method in 3.2, and finally cut the parameter increment of new tasks to be orthogonal to historical ones in 3.3.

### 3.1 MOTIVATION: ANALYSIS OF EXISTING REGULARIZATION METHODS

Our method is developed using a regularization-based approach in consideration of its little computational or memory overhead issues (Wu et al., 2024b). While prior research Zheng et al. (2025) has demonstrated that existing regularization methods are efficient on short task sequences, their performance is relatively limited in long sequences, leaving a gap for their practical deployment. In the following parts, we propose an analysis to identify the primary cause of this defect.

**Intrinsic functional directions of LLMs**. Functional directions (Olah et al., 2020; 2018; Saxena & Cunningham, 2019) have become prevalent in research on LLMs. In this paper, we define functional directions of LLMs as *the gradient direction of model parameters on certain datapoints (batches)*. Most of the prior regularization-based approaches employ functional directions to approximate the functional unit of certain tasks in LLMs. They adhere to the outline for recording functional directions and regularizing new updates on historical directions. Specifically, Orthogonal methods, including Orthogonal Gradient Descent (OGD) (Farajtabar et al., 2020) and Orthogonal Subspace Learning (O-LoRA) (Wang et al., 2023a), avoid perturbing the historical settings of the model

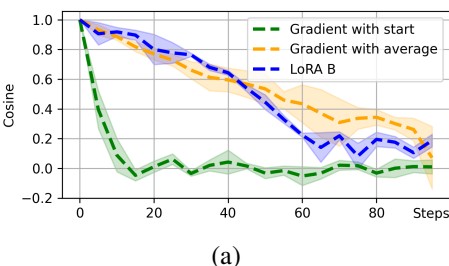 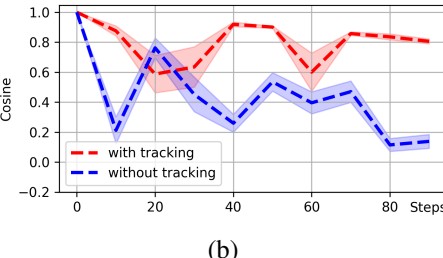

(a)                                                         (b)

Figure 2: Quantification of functional direction drift regarding a particular datapoint (batch) $(x, y)$. Figure (a) shows the cosine similarity between current and historical functional directions. The green line shows $\mathbf{cos}\langle G_T, G_1 \rangle$, where $G_T = \nabla_\theta L(F_{\theta_T}(x), y)$, $\theta_T$ is the model parameter in the $T$ th fine-tuning step. The yellow line shows $\mathbf{cos}\langle G_T, \bar{G}_T \rangle$, where $\bar{G}_T = \frac{1}{T}\sum_{t=1}^{T} G_t$. The blue line shows the average similarity of $\beta_1, \beta_2, ..., \beta_r$ in LoRA $B$ matrices with their start value, *i.e.* $\frac{1}{r}\sum_{n=1}^{r}\mathbf{cos}\langle \tilde{\beta}_n, \beta_n \rangle$, where $\tilde{\beta}$ is the current one, $\beta$ is the start one. Please refer to Figure 5 in the Appendix for more results. Figure (b) shows the effect of tracking functional directions. We initialize the principal components during the first dataset, and measure the drift in the following steps. The red line shows the drift with $\mathbf{cos}\langle \mathbf{coord}(h_T^*), \mathbf{coord}(h_1^*) \rangle$, where $h_T^*$ is the LoRA increment (computed with Equation (5)) in the $T$-th step. For contradiction, we freeze the update of principal components (the blue line). The results are the average of randomly-chosen datapoints, with standard deviation shown.

through orthogonal approaches, and the two respectively employ gradient directions and LoRA vectors as the regularized functional directions. Elastic Weight Consolidation(EWC) methods (Kirkpatrick et al., 2017; Zenke et al., 2017) employ the Fisher information matrix for its consolidation, which is also computed with gradients.

**Functional directions drift in the fine-tuning process**. In this part, we identify that the drift of functional directions during the continuous fine-tuning process is the key issue of the regularizations above. Specifically, in the process of continually fine-tuning an LLM, the locality of linearity in its deep neural networks is broken (Black et al., 2022), thus destroying the functionality of the directions extracted in earlier steps. Consequently, regularization in these directions deviates from the original purpose in the continual fine-tuning, as demonstrated in Figure 1(a). In this part, we present the following observations regarding the drifts proposed above. We take fine-tuning LLaMA-7B (Touvron et al., 2023) on CL Benchmark (Zhang et al., 2015) as the example in this experiment, and measure the drift of the gradient direction during continual fine-tuning.

As shown in Figure 2(a), with the fine-tuning process conducted, the functional directions captured earlier no longer represent the current ones, exposing the ineffectiveness of employing *fixed* singular or average gradient as the functional direction, which is conducted in EWC (Kirkpatrick et al., 2017; Zenke et al., 2017) and OGD (Farajtabar et al., 2020). Similarly, we also investigate the drift of column vectors in LoRA $B$ matrices employed by O-LoRA (Wang et al., 2023a), *i.e.* $\beta_1, \beta_2, ..., \beta_r$ in $B = (\beta_1, \beta_2, ..., \beta_r)$. The results (blue line, denoted as LoRA $B$) show that employing column vectors in LoRA $B$ matrices mitigates the loss of functional directions. However, it does not resolve the drift issue fundamentally. Overall, we identify that the prevalent problem in prior regularization-based methods is the drift of functional directions, indicating that we need to dynamically update the functional directions rather than relying on fixed ones.

### 3.2 TRACKING THE DRIFT OF FUNCTIONAL DIRECTIONS

The difficulty of mitigating the drift of functional directions lies in updating the functional directions of historical data in the current parameter space, as there is no access to historical data in the settings of regularization-based methods. To tackle this issue, we propose our method to track the drifting functional directions of historical tasks with the latest task data.

As shown in Equation (3), LLMs primarily fine-tune within a low-rank subspace, *i.e.* the space of $BAx$, such that all tasks share most of the bases in this subspace, and the functional directions are different linear combinations of these bases. By extracting and updating the bases from the functional directions of the current task, we update the shared bases of historical functional directions, thus updating historical functional directions themselves.

**Tracking method overview**. To achieve the conception above, we select the LoRA increment as the functional direction, and employ Principal Component Analysis (PCA) to extract the bases. The following parts propose respective elaborations.

**LoRA increment as functional directions**. Following prior regularization-based methods, including OGD(Farajtabar et al., 2020) and O-LoRA(Wang et al., 2023a), we extract fine-tuning increments as the functional directions of certain continual learning tasks. More specifically, the functional direction we select is the increment of LoRA in Equation (3), that is:

$$\mathbf{d}(W_m x_m) = \mathbf{d}(B_m A_m x_m) = \mathbf{d}(B_m A_m) x_m = (\mathbf{d}B_m) A_m x_m + B_m (\mathbf{d}A_m) x_m \triangleq p_m, \quad (4)$$

where $x_m$ is the input *vector* to the $m$-th LoRA module with parameter $B_m$ and $A_m$. Let $\alpha$ be the learning rate, then $\mathbf{d}B = \alpha \nabla_B L$, $\mathbf{d}A = \alpha \nabla_A L$, $L$ is the task-specific loss function. We represent the update direction of LoRA with the following concatenated vector:

$$h = \mathbf{concat}(p_1, p_2, ..., p_M), \quad (5)$$

where $M$ is the number of LoRA modules. The concatenation captures the relation between the LoRA increment of different layers. More computational details on $x$ and $h$ are in Appendix D.

**Online PCA**. To extract the basis of functional directions from their linear combinations, we employ the Online Principal Component Analysis (Online PCA) (Cardot & Degras, 2018), which requires only the latest data in memory, conforming to the settings of regularization-based continual learning.

The target of Online PCA is as follows. Let $\{h_1, h_2, ..., h_n\}$ be functional directions computed with Equation (5) on a sequence of incoming data. On receiving a new functional direction $h_t$, Online PCA seeks to update principal components $\{v_t^1, v_t^2, ..., v_t^{K_t}\}$ as the basis of functional directions $\{h_1, h_2, ..., h_t\}$. Moreover, when processing the latest data $h_t$, there is no access to historical datas $\{h_1, h_2, ..., h_{t-1}\}$. This realizes our goal of updating historical functional directions with current ones, *i.e.* updating the representation of historical functional directions with the current functional direction. Please refer to Algorithm 1 for a summary and Figure 1(a) for a brief demonstration.

There are multiple approaches to implement Online PCA, including Incremental PCA (Arora et al., 2012; Levy & Lindenbaum, 1998) and stochastic approximation methods (Sanger, 1989; Krasulina, 1970; Oja & Karhunen, 1985; Oja, 1992). Our method draws inspiration from Candid Covariance-free Incremental PCA (CCIPCA) (Weng et al., 2003), since its edge lies in the ability to add components freely, which is suited for emerging new tasks. It also has a lower computational overhead compared to other techniques (Cardot & Degras, 2018). Please refer to Appendix A for more technical details on our Online PCA method. To further evaluate the above Online PCA techniques, we present evaluation experiments in Section 4.3.

**The effectiveness of tracking**. To evaluate the effectiveness of tracking, we investigate the drift of the functional direction in the subspace of the updated principal components. Specifically, we compute the LoRA increment $h_T^*$ of a particular datapoint in the $T$-th fine-tuning step, and compute its coordinate in the subspace of extracted principal components, that is

$$\mathbf{coord}(h_T^*) = \left( (h_T^*, v_T^1), (h_T^*, v_T^2), ..., (h_T^*, v_T^K) \right). \quad (6)$$

where $(h_T^*, v_T^k) = \frac{h_T^* \cdot v_T^k}{\|v_T^k\|}$ is the projection of $h_T^*$ on $v_T^k$. As shown in Figure 2 (b), by tracking principal components, drifting functional directions can be followed and thus remain in correspondence with their original states; if we forbid tracking, functional directions are gradually lost.

### 3.3 CUT FINE-TUNING DIRECTIONS FOR FUNCTION PRESERVATION

Following prior orthogonal space fine-tuning approaches, including OGD (Farajtabar et al., 2020) and O-LoRA (Wang et al., 2023a), for regularization-based continual learning, we try to make the parameter increment of new tasks orthogonal to historical ones. This avoids changing the functional directions representing historical tasks, thus protecting historical functions. Specifically, we make the current LoRA increment $h_T$ orthogonal to historical ones, whose basis are principal components $\{v_T^1, v_T^2, ..., v_T^K\}$. The goal is as follows:

$$h_T \perp v_T^k \quad k = 1, 2, ..., K. \quad (7)$$

Table 1: Average Accuracy (AA) of different continual methods on LLaMA-7B, including DOC with **(w.)** different online learning techniques.

| | | Standard CL Benchmark | | | | Long chain of tasks | | | |
| --- | --- | --- | --- | --- | --- | --- | --- | --- | --- |
| | | Order 1 | Order 2 | Order 3 | **Average** ($\uparrow$) | Order 4 | Order 5 | Order 6 | **Average** ($\uparrow$) |
| **Main comparison** | LoRA | 67.7 | 65.4 | 66.2 | 66.4 | 61.2 | 63.6 | 60.7 | 61.8 |
| | EWC | 72.3 | 65.0 | 70.4 | 69.2 | 59.7 | 61.2 | 65.4 | 62.1 |
| | LwF | 71.6 | 66.0 | 69.7 | 69.1 | 60.8 | 62.6 | 63.3 | 62.2 |
| | O-LoRA | 78.2 | 76.4 | 74.7 | 76.5 | **71.7** | 73.8 | 70.2 | 71.9 |
| | **DOC**-main | **80.5** | 78.6 | 73.9 | **77.7** | 71.6 | **74.1** | **74.4** | **73.4** |
| **Ablation study** | **DOC** w. IPCA | 77.6 | **79.4** | 74.7 | 77.2 | 69.7 | 72.6 | 73.8 | 72.0 |
| | **DOC** w. SGA | 78.2 | **79.4** | **75.2** | 77.6 | 70.4 | 73.0 | 69.9 | 71.1 |
| | **DOC** w. GHA | 79.3 | 77.2 | 74.9 | 77.1 | 69.8 | 70.5 | 68.4 | 69.6 |
| | **DOC** w. SNL | 79.8 | 78.5 | 74.5 | 77.6 | 71.1 | 72.1 | 68.5 | 70.5 |
| | **DOC**-strong | 78.0 | 75.7 | 75.1 | 76.2 | 70.0 | 67.7 | 68.3 | 68.6 |
| | **DOC**-weak | 70.2 | 69.5 | 71.6 | 70.4 | 63.4 | 64.6 | 64.6 | 64.2 |
| | **DOC**-freeze | 70.7 | 69.5 | 67.3 | 69.1 | 60.0 | 62.5 | 64.9 | 62.4 |
| **Oracle methods** | Replay | 67.9 | 68.2 | 71.0 | 69.0 | 62.3 | 65.0 | 61.4 | 62.9 |
| | PerTaskLoRA | 76.9 | 76.9 | 76.9 | 76.9 | 76.8 | 76.8 | 76.8 | 76.8 |
| | MTL | 83.4 | 83.4 | 83.4 | 83.4 | 80.3 | 80.3 | 80.3 | 80.3 |
| | ProgPrompt | 77.4 | 76.9 | 77.9 | 77.4 | 76.8 | 76.2 | 77.1 | 76.7 |

---

**Algorithm 1** DOC (Our method)

---

**Input**: Model $F_\theta$, where $\theta = (A, B)$ includes LoRA $A$, $B$ modules; learning rate $\alpha$; the $t$-th incoming dataset $D_t$, expected maximum principal component number $K$ for each new task

**Initialization**: Principal components $v_T^1, v_T^2, ..., v_T^{K_T}$ extracted from historical fine-tunings, $T$ is the number of finished fine-tuning steps.

**Output**: Fine-tuned parameter $\theta^*$

1: **for** data point(batch) $(x_t^i, y_t^i)$ in $D_t$ **do**
2:      extract gradients: $\nabla_B L = \nabla_B L(F_\theta(x_t^i), y_t^i) \quad \nabla_A L = \nabla_A L(F_\theta(x_t^i), y_t^i)$
3:      compute current LoRA increment $h_{T+i}$ with Equation (5)
4:      use $h_{T+i}$ to update principal components with Online PCA Algorithm on the basis of existing $v_{T+i-1}^1, v_{T+i-1}^2, ..., v_{T+i-1}^{K_{T+i-1}}$, get $v_{T+i}^1, v_{T+i}^2, ..., v_{T+i}^{K_{T+i}}$ $(K_{T+i-1} \le K_{T+i} \le K \cdot t)$
5:      cut $\nabla_B L$ with Equation (13), get $(\nabla_B L)_{\text{cut}}$
6:      update parameter: $B = B - \alpha \cdot (\nabla_B L)_{\text{cut}} \quad A = A - \alpha \cdot \nabla_A L$
7: **end for**
8: **return** $\theta^* = (A, B)$

---

Note that in Equation (5) we have

$$h_T = \textbf{concat}\left(\mathbf{d}(B_m A_m x_m) \quad m = 1, 2, ..., M\right), \tag{8}$$

so we disassemble the concatenation to realize the orthogonality in Equation (7) as follows:

$$v_T^k = \textbf{concat}(v_T^k(m) \quad m = 1, 2, ..., M). \tag{9}$$

Then we only need to make

$$\mathbf{d}(B_m A_m x_m) \perp v_T^k(m) \quad m = 1, 2, ...M \quad k = 1, 2, ..., K \tag{10}$$

**Note that we substitute $BAx$ for $B_m A_m x_m$ and $\tilde{v}^k$ for $v_T^k(m)$ in the following statements for concision**. As $\mathbf{d}(BAx) = (\mathbf{d}B)Ax + B(\mathbf{d}A)x$, we realize the orthogonality in Equation (10) respectively for $(\mathbf{d}B)Ax$ and $B(\mathbf{d}A)x$:

$$(\mathbf{d}B)Ax \perp \tilde{v}^k, \quad B(\mathbf{d}A)x \perp \tilde{v}^k \quad \text{for} \quad k = 1, 2, ..., K.$$

For $(\mathbf{d}B)Ax$, note that

$$(\mathbf{d}B)Ax = (\mathbf{d}\beta_1, \mathbf{d}\beta_2, ..., \mathbf{d}\beta_r)(Ax) \in \langle \mathbf{d}\beta_1, \mathbf{d}\beta_2, ..., \mathbf{d}\beta_r \rangle \tag{11}$$

So we only need to cut $\mathbf{d}\beta_i = \alpha \cdot \nabla_{\beta_i} L \quad (i = 1, 2, .., r)$ to be orthogonal to $\tilde{v}^k \quad (k = 1, 2, ..., K)$:

$$\nabla_{\beta_i} L \perp \tilde{v}^k \quad i = 1, 2, ..., r \quad k = 1, 2, ..., K \tag{12}$$

Then we reach the following gradient cut:

$$(\nabla_{\beta_i} L)^* = \nabla_{\beta_i} L - \sum_{k=1}^{K} \frac{\nabla_{\beta_i} L \cdot v_T^k}{\|v_T^k\|^2} \cdot v_T^k \quad i = 1, 2, ..., r. \tag{13}$$

Now we get $(\nabla_B L)_{\text{cut}} = ((\nabla_{\beta_1} L)^*, (\nabla_{\beta_2} L)^*, ..., (\nabla_{\beta_r} L)^*)$. Note that the cut above removes the correlation with input $x$ since Equation (11), making the orthogonality hold true for all kinds of input $x$. This is significant in preserving historical functions on all tasks.

For $B(\mathbf{d}A)x$, assume we have employed $(\nabla_B L)_{\text{cut}}$ in previous steps, then their aggregated $B = (\beta_1, \beta_2, ..., \beta_r)$ satisfies the orthogonality for the former steps. Similar to Equation (11), we have

$$B(\mathbf{d}A)x \in \langle \beta_1, \beta_2, ..., \beta_r \rangle, \tag{14}$$

so the orthogonality holds for $B(\mathbf{d}A)x$. We keep $B(\mathbf{d}A)x$ intact as a momentum for optimization, which means we keep the original $\mathbf{d}A$ and $\nabla_A L$.

Please note that the above orthogonal cut does not harm the gradient descent, as described in the paper of OGD (Farajtabar et al., 2020). In summary, our complete method is formulated as Algorithm 1. Please refer to Figure 1 (b) for a brief demonstration.

Table 2: Average Accuracy (AA) of different continual methods on LLaMA-13B.

| | | Standard CL Benchmark | | | | Long chain of tasks | | | |
|---|---|---|---|---|---|---|---|---|---|
| | | Order 1 | Order 2 | Order 3 | **Average** (↑) | Order 4 | Order 5 | Order 6 | **Average** (↑) |
| **Main comparison** | LoRA | 69.2 | 68.0 | 65.7 | 67.6 | 59.9 | 64.7 | 62.0 | 62.2 |
| | EWC | 72.7 | 66.9 | 66.0 | 68.5 | 63.4 | 60.2 | 66.7 | 63.4 |
| | LwF | 71.0 | 70.4 | 72.8 | 71.4 | 64.5 | 62.6 | 65.3 | 64.1 |
| | O-LoRA | 77.9 | 79.8 | 77.6 | 78.4 | 70.8 | 73.2 | 72.2 | 72.0 |
| | **DOC**-main | **79.5** | **81.2** | **79.7** | **80.1** | **72.4** | **74.0** | **76.5** | **74.3** |
| | **DOC**-freeze | 69.0 | 74.6 | 70.9 | 71.5 | 62.6 | 62.3 | 66.0 | 63.6 |
| **Oracle methods** | Replay | 70.1 | 69.4 | 68.2 | 69.2 | 64.3 | 65.4 | 63.6 | 64.4 |
| | PerTaskLoRA | 77.4 | 77.4 | 77.4 | 77.4 | 78.5 | 78.5 | 78.5 | 78.5 |
| | MTL | 85.7 | 85.7 | 85.7 | 85.7 | 83.6 | 83.6 | 83.6 | 83.6 |
| | ProgPrompt | 76.2 | 80.9 | 78.5 | 78.5 | 79.9 | 80.0 | 78.0 | 79.3 |

Table 3: Average BWT and FWT scores of different continual methods on LLaMA-7B, including DOC with (w.) different Online PCA techniques

| | | Standard CL Benchmark | | Long chain of tasks | |
|---|---|---|---|---|---|
| | | BWT(↑) | FWT(↑) | BWT | FWT |
| **Main comparison** | LoRA | $-14.6_{+0.0}$ | $0.6_{+0.0}$ | $-16.2_{+0.0}$ | $0.2_{+0.0}$ |
| | EWC | $-10.6_{+4.0}$ | $0.2_{-0.4}$ | $-14.3_{+1.9}$ | $-1.5_{-1.7}$ |
| | LwF | $-10.9_{+3.7}$ | $0.5_{-0.1}$ | $-15.0_{+1.2}$ | $-0.6_{-0.8}$ |
| | O-LoRA | $-1.9_{+12.7}$ | $1.4_{+0.8}$ | $-5.2_{+11.0}$ | $0_{-0.2}$ |
| | **DOC**-main | $\mathbf{-0.6_{+14.0}}$ | $\mathbf{1.6_{+1.0}}$ | $\mathbf{-3.4_{+12.8}}$ | $-0.1_{-0.3}$ |
| **Ablation study** | **DOC** w. IPCA | $-0.8_{+13.8}$ | $1.2_{+0.6}$ | $-4.8_{+11.4}$ | $-0.3_{-0.5}$ |
| | **DOC** w. SGA | $-0.7_{+13.9}$ | $\mathbf{1.6_{+1.0}}$ | $-5.6_{+10.6}$ | $-0.5_{-0.7}$ |
| | **DOC** w. GHA | $-1.0_{+13.6}$ | $1.2_{+0.6}$ | $-7.9_{+8.3}$ | $\mathbf{0.2_{+0.0}}$ |
| | **DOC** w. SNL | $-0.7_{+13.9}$ | $1.6_{+1.0}$ | $-6.0_{+10.2}$ | $-0.7_{-0.9}$ |
| | **DOC**-strong | $-1.9_{+12.7}$ | $1.0_{+0.4}$ | $-7.7_{+8.5}$ | $-1.1_{-1.3}$ |
| | **DOC**-weak | $-7.8_{+6.8}$ | $-0.3_{-0.9}$ | $-13.6_{+2.6}$ | $0.1_{-0.1}$ |
| | **DOC**-freeze | $-8.8_{+5.8}$ | $-1.5_{-2.1}$ | $-13.7_{+2.5}$ | $-1.7_{-1.9}$ |
| **Oracle methods** | Replay | $-10.5_{+4.1}$ | $0_{-0.6}$ | $-14.7_{+1.5}$ | $0.2_{+0.0}$ |
| | ProgPrompt | $-0.2_{+14.4}$ | $0.8_{+0.2}$ | $-0.2_{+16.0}$ | $0.1_{-0.1}$ |

## 4 EXPERIMENTS

In this section, we test our method across various LLM continual learning benchmarks through extensive experiments to explore the practical impact on real-world continual deployment with online streaming data.

## 4.1 SETUP

**Datasets and Models**. Following ProgPrompt (Razdaibiedina et al., 2023) and O-LoRA (Wang et al., 2023a), we employ CLBenchmark (AG News, Amazon reviews, Yelp reviews, DBpedia, Yahoo answers) (Zhang et al., 2015) to evaluate our methods, adding GLUE (MNLI, QQP, RTE, SST2) (Wang et al., 2018), SuperGLUE (WiC, CB, COPA, MultiRC, BoolQ) (Wang et al., 2019), and IMDB review (Maas et al., 2011) for long-chain tasks. The models we use are LLaMA-7B, LLaMA-13B (Touvron et al., 2023), and T5-Large (Colin, 2020).

**Metrics**. Following ProgPrompt and O-LoRA, we employ **Average Accuracy (AA)** to evaluate the overall performance of continual learning, that is $\mathbf{AA}(T) = \frac{1}{T} \sum_{t=1}^{T} a_{t,T}$ where $a_{t,T}$ is the test accuracy on the $t$-th task after fine-tuning on the $T$-th task.

In order to measure the catastrophic forgetting, we employ Backward Transfer Rate (BWT) and Forward Transfer Rate (FWT) (Wu et al., 2022):

$$\mathbf{BWT}(T) = \frac{1}{T-1} \sum_{t=1}^{T-1} (a_{t,T} - a_{t,t}), \quad \mathbf{FWT}(T) = \frac{1}{T-1} \sum_{t=2}^{T} (a_{t,t} - \tilde{a}_t). \quad (15)$$

Commonly, in a continual learning scenario, a negative BWT score indicates forgetting, and a negative FWT reveals that we regularize the fine-tuning process and decrease the fine-tuning performance $a_{t,t}$ compared to a standard fine-tuning performance $\tilde{a}_t$.

Overall, a regularization-based method pursues a higher BWT score meaning less forgetting, at the cost of a smaller decrease in FWT score, representing less damage to the fine-tuning of each task.

**Implementation details**. For LLaMA-7B and LLaMA-13B, we set the learning rate $\alpha = 1e-4$, with a batch size of 8. For T5-Large, we let $\alpha = 1e-3$ with a batch size of 64, following O-LoRA. Please refer to Appendix C for more details, including step number, task sequence, instructions, *etc.*

**Compared methods**. To ensure a fair comparison, we primarily focus on the recent **state-of-the-art regulation-based methods**, including EWC, LwF, and O-LoRA. We also consider fine-tuning the model with task-specific datasets sequentially using LoRA (Hu et al., 2022), which is a vanilla baseline and the expected lower bound of continual learning. Note that other-based methods require additional settings and are **not comparable** to ours, which is detailed in Appendix E. Furthermore, we present the results from several other oracle methods that are not suitable for continuous fine-tuning settings, but they can serve as upper bounds:

- **Replay** replay samples from historical tasks when fine-tuning on new tasks.
- **PerTaskLoRA** train LoRA modules solely for each task.
- **MTL** train the model on all tasks as multi-task learning.
- **ProgPrompt** (Razdaibiedina et al., 2023) a state-of-the-art method that updates an extending prompt in the streaming data, but task ID is required during inference.

Table 4: Average Accuracy (AA) of different continual methods on T5-large

| | | Standard CL Benchmark | | | | Long chain of tasks | | | |
| --- | --- | --- | --- | --- | --- | --- | --- | --- | --- |
| | | Order 1 | Order 2 | Order 3 | Average (↑) | Order 4 | Order 5 | Order 6 | Average (↑) |
| **Main comparison** | LoRA | 44.6 | 32.7 | 53.7 | 43.7 | 2.3 | 0.6 | 1.9 | 1.6 |
| | EWC | 48.7 | 47.7 | 54.5 | 50.3 | 45.3 | 44.5 | 45.6 | 45.1 |
| | LwF | 54.4 | 53.1 | 49.6 | 52.3 | 50.1 | 43.1 | 47.4 | 46.9 |
| | O-LoRA | 75.4 | 75.7 | **76.3** | 75.8 | 72.3 | 64.8 | 71.6 | 69.6 |
| | **DOC** (ours) | **78.8** | **78.8** | 74.5 | **77.4** | 72.7 | **72.4** | **74.0** | **73.0** |
| | **DOC**-freeze | 62.1 | 62.9 | 60.4 | 61.8 | 55.6 | 52.5 | 57.7 | 55.3 |
| **Oracle methods** | Replay | 55.2 | 56.9 | 61.3 | 57.8 | 55.0 | 54.6 | 53.1 | 54.2 |
| | PerTaskLoRA | 70.0 | 70.0 | 70.0 | 70.0 | 78.1 | 78.1 | 78.1 | 78.1 |
| | MTL | 80.0 | 80.0 | 80.0 | 80.0 | 76.5 | 76.5 | 76.5 | 76.5 |
| | ProgPrompt | 75.2 | 75 | 75.1 | 75.1 | 78.0 | 77.7 | 77.9 | 77.9 |

## 4.2 MAIN RESULTS

Following ProgPrompt and O-LoRA, there are three independent runs with different task orders for different chains of tasks, as detailed in Appendix C.

Table 5: (a) Average clock time of one fine-tuning step; (b) Average Accuracy (AA) results of standard CL Benchmark on LLaMA-7B with different LoRA rank $r$ and maximum principal component number $K$ for each new task. The results are the average of task orders 1-3.

| (a) | | | | (b) | | | | | |
|---|---|---|---|---|---|---|---|---|---|
| | LLaMA-7B | LLaMA-13B | T5-Large | | $K$ | 32 | 48 | 64 | 96 |
| LoRA | 0.38s | 0.97s | 0.68s | **DOC** | $r=16$ | 76.1 | 77.6 | 76.5 | 76.5 |
| | | | | -main | $r=64$ | 77.9 | 78.5 | 78.4 | **78.7** |
| EWC | 0.42s | 1.20s | 0.76s | LoRA | $r=16$ | | 65.4 | | |
| LwF | 0.40s | 1.23s | 0.76s | | $r=64$ | | 67.4 | | |
| O-LoRA | 0.42s | 1.29s | 0.78s | O-LoRA | $r=16$ | | 77.0 | | |
| DOC-main | 0.46s | 1.23s | 0.80s | | $r=64$ | | 76.8 | | |
| ProgPrompt | 0.32s | 0.43s | 0.31s | | | | | | |

Table 6: Average clock time per fine-tuning step (update over one batch) of DOC with **(w.)** different Online PCA techniques. The model is LLaMA-7B.

| | DOC | DOC w. IPCA | DOC w. SGA | DOC w. GHA | DOC w. SNL |
|---|---|---|---|---|---|
| clocktime | 0.46s | 0.92s | 0.88s | 0.67s | 0.66s |

**Overall performance**. The results of Average Accuracy (AA) are shown in Table 1, Table 2, and Table 4. We refer to the paper of O-LoRA (Wang et al., 2023a), the up-to-date regularization-based method, for the results of other approaches on T5-Large, as the settings and hyperparameters of our experiments are equal. The results show that our method outperforms prior ones, especially in long-chain tasks. We respectively achieve an accuracy of 77.4 and 73.0 in the standard CL benchmark and long chains of tasks for LLaMA-7B, compared to 75.8 and 69.6 for O-LoRA, the previous state-of-the-art regularization-based method.

**Mitigating Catastrophic Forgetting**. The BWT and FWT results are shown in Table 3. The BWT score of our method is higher than that of prior approaches. We reach -0.6 and -3.4 for standard and long continual learning tasks, compared to -1.9 and -5.2 of O-LoRA, indicating that our method suffers less from catastrophic forgetting. Our FWT score, compared to other methods, indicates that we mitigate catastrophic forgetting at a slight cost to fine-tuning performance.

In summary, our method mitigates forgetting with a much higher BWT score, at the cost of a little fine-tuning performance with a slightly lower FWT score, eventually reaching effective overall performances and higher AA scores.

## 4.3 DISCUSSIONS

**The choice of different Online PCA approaches**. We evaluate employing different Online PCA techniques in our method using the CL Benchmark on LLaMA-7B. Specifically, the following techniques are involved:

- Incremental PCA (IPCA) (Arora et al., 2012): a common technique based on the incremental SVD.
- Stochastic Gradient Ascent (SGA) (Oja, 1992) and Generalized Hebbian Algorithm (GHA) (Sanger, 1989): update principal components via optimization.
- Subspace Network Learning (SNL) (Oja, 1992): also optimization-driven, faster than SGA, but only covers the principal component subspace rather than exact vectors.

Note that these techniques are not capable of adding principal components freely, so we update $N$ components from the beginning ($N = 200$ for CL Banchmark and $N = 400$ for long chain of tasks), and involve more components in the computation when new tasks come, to simulate adding components.

We evaluate the tracking quality of different Online PCA techniques in Figure 3 (a). The continual learning results are shown in Table 1 and Table 3, and their computational costs are in Table 6. Results show that changing the Online PCA technique in our method does not fundamentally change the tracking ability and continual learning performance, which generally verifies our perspective of

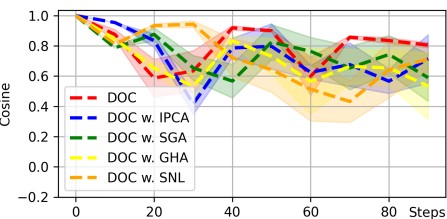 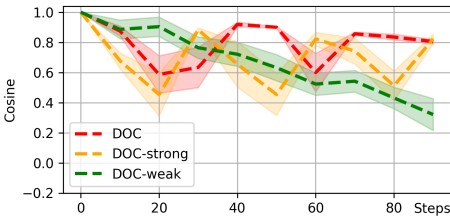

(a) DOC with Different Online PCA     (b) DOC with different tracking strength

Figure 3: The tracking effect of different Online PCA techniques and tracking strengths. We plot $\cos\langle \mathbf{coord}(h_T^*), \mathbf{coord}(h_1^*) \rangle$ as described under Figure 2. The results are the average of random data points in the CL Benchmark on LLaMA-7B, with standard deviation shown. Figure (a) shows that changing the PCA technique does not fundamentally affect tracking performance, with our method using CCIPCA (red line) performing slightly better; Figure (b) shows that a moderate tracking strength is more stable compared to a strong one and more effective compared to a weak one.

tracking functional directions. But please note that the CCIPCA is still an optimal choice because of its ability to freely add new components and its light computational overhead.

**Computational costs**. The cost of storing all principal components (with maximum principal component number $K \leq 100$) is within 100MB, roughly equivalent to a few sets of LoRA modules, and is negligible compared to the cost of fine-tuning the model itself. We employ vGPU-48GB as our device, with PyTorch 2.1.0 and CUDA 12.1, and the clock time of one training step with different regularization methods is shown in Table 5. As the Online PCA technique we employ has an explicit update expression (shown in Appendix A), it does not incur much extra computational costs.

**The choice of hyperparameters**. We present the following empirical study regarding different choices of LoRA rank, say $r$, for fine-tuning, and the maximum principal component number for each new task, say $K$, for functional direction tracking. The results are shown in Table 5. Overall, adequate principal components cooperating with higher LoRA ranks are able to cover and protect more critical functional directions for historical tasks, thus ensuring a better historical functional preservation and task accuracy. The results show that the variation between different choices of hyperparameters is little, revealing the robustness of our method. The CCIPCA technique also introduces other hyperparameters, please refer to Appendix C for further description.

**Ablation study**. We conduct a trial on freezing the update of principal components to investigate the impact of functional direction tracking. Specifically, we cease updating principal components after their initialization during the first 10% fine-tuning steps for each task. The results are shown in the **DOC-freeze** line in Table 1, 2, 3, and 4. The decrease in continual learning performance in the ablation experiment indicates that it is tracking the functional directions that mitigate catastrophic forgetting and enhance the performance of continual learning.

For further demonstration, we vary the tracking strengths and investigate the subsequent tracking quality and continual learning accuracy. Specifically, we strengthen or weaken the update of the principal component, get **DOC-strong** and **DOC-weak**, with their technical settings elaborated under Table 7 in the Appendix. The subsequent tracking qualities are shown in Figure 3. The results show that moderate tracking strength is more stable compared to a strong one and more effective compared to a weak one. Performances of continual learning are shown in Table 1 and Table 3, line **DOC-strong** and **DOC-weak**. Since a moderate tracking strength makes an optimal tracking quality, its continual learning performance is better, consequently.

## 5   CONCLUSION

In this paper, we introduce a novel regularization-based approach that leverages functional direction tracking for continual learning in language models. We identify that the drift of functional directions is the key issue for regularization-based continual learning approaches, and the proposed method systematically addresses the drift issue by updating the functional directions dynamically with Online PCA during the fine-tuning process. Empirical evaluations verify the effectiveness of our tracking method and underscore its efficacy in enhancing continual learning performance. For limitations and future directions, please refer to Appendix F and G.

ETHICS STATEMENT

This work focuses on developing fine-tuning methods to mitigate catastrophic forgetting in LLMs for continual learning, with no involvement of human subjects, sensitive personal data, or high-risk real-world deployments.

REPRODUCIBILITY STATEMENT

Our code will be available upon publication. All datasets and LLMs we used in experiments are publicly available online.

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

# A  ONLINE PCA IN OUR METHOD

In our method, we extract the basis of functional directions from their linear combinations using Online PCA (Cardot & Degras, 2018). Specifically, we employ a modified Candid Covariance-free Incremental PCA (CCIPCA) (Weng et al., 2003) to implement Online PCA.

**The CCIPCA technique**. Let $\Gamma = \frac{1}{T-1} HH^\top$ be the covariance matrix $H = (h_1, h_2, ..., h_N)$ with standardized datas $h_1, h_2, ..., h_N$. Recall the goal of the PCA is to find the eigenvector $u$ and the eigenvalue $\lambda$ of $\Gamma$ that satisfy

$$\Gamma u = \lambda u. \tag{16}$$

The idea of CCIPCA is as follows. For the first eigenvector $v^1$, assume that estimates $v^1_0, ..., v^1_{T-1}$ of $v = \lambda u$ have been constructed in previous steps $t = 1, 2, ..., T-1$. We substitute $h_t h_t^\top$ to $\Gamma$ and $\frac{v^1_{t-1}}{\|v^1_{t-1}\|}$ to $u$ in the eigenequation (16) for $t = 1, ..., T$, and average the results:

$$v^1_T = \frac{1}{T} \sum_{t=1}^{T} h_t h_t^\top \frac{v^1_{t-1}}{\|v^1_{t-1}\|}. \tag{17}$$

Note that CCIPCA requires no historical datas $\{h_1, h_2, ..., h_{T-1}\}$, as equation (17) can be conveniently written in recursive form as:

$$v^1_{T+1} = \frac{T-l}{T+1} v^1_T + \frac{1+l}{T+1} h_{T+1} h_{T+1}^\top \frac{v^1_T}{\|v^1_T\|}, \tag{18}$$

where an amnesic factor $l \geq 0$ is introduced to handle nonstationary data generation, and the initialization is $v_0 = h_1$. The almost-sure convergence of equation (18) has been proved by (Zhang & Weng, 2001). For estimating more than one eigenvector, say $v^1, v^2, ..., v^K$, to update the $K$-th eigenvector $v^K_{T+1}$, simply replace the input vector $h_{T+1}$ in equation (18) with the following residual cutting:

$$h^*_{T+1} = h_{T+1} - \sum_{k=1}^{K-1} \frac{h_{T+1} \cdot v^k_T}{\|v^k_T\|^2} \cdot v^k_T. \tag{19}$$

**Modified CCIPCA for tracking functional directions**. To deal with the issue of functional direction drift, we introduce a tracking factor $\epsilon \in (0, \epsilon_{\max})$ to equation (18) for a faster update:

$$v^1_{T+1} = \eta \cdot v^1_T + (1 - \eta) \cdot h_{T+1} h_{T+1}^\top \frac{v^1_T}{\|v^1_T\|}, \tag{20}$$

where $\eta = \frac{T-l}{T+1} \cdot (1 - \epsilon)$. Note that the convergence of equation (20) is disturbed for tracking. Algorithm 2 summarizes the modified tracking CCIPCA method, which additionally employs a residual threshold $\delta \in (0, 1)$ to append new components automatically (lines 7-10).

**An example of functional direction tracking**. We present the following example on the working process of functional direction tracking as a reference for Algorithm 2. Still, we take fine-tuning Llama-2 on CLBenchmark as an example. As shown in Figure 4, we update the principal components based on the residual rate $\frac{\|h^*_t\|}{\|h_t\|}$ with residual threshold $\delta$, abiding by lines 7-10 in Algorithm 2. Note that a lower residual rate indicates more complete coverage of LoRA increment with existing principal components. The tracking factor $\epsilon$ is adjusted dynamically following the increase and decrease of the residual rate, which is executed by redoing lines 2-6 in Algorithm 2 with adjusted $\epsilon$ and $\eta$. The results show that we continuously keep the residual rate less than 10%, covering 90% of the LoRA increment.

# B  MORE EXPERIMENTAL RESULTS AS MOTIVATION

In Section 3.1, we denote that the drift of functional directions in the fine-tuning process is the key issue lying in existing regularization-based continual learning methods, and demonstrate the drift in Figure 2. To further show the general existence of functional direction drift, we present the following demonstration using the entire CL Benchmark dataset on LLaMA-7B. Specifically, we measure the

---

**Algorithm 2** Online PCA for Tracking Functional Directions

---

**Parameter**: Maximum principal component number $K_{\max}$, amnesic factor $l$, tracking factor $\epsilon$, residual threshold $\delta$
**Initialization**: current principal component number $n = 0$
**Input**: The incoming model state data $h_{T+1}$
**Output**: Updated principal components $v_{T+1}^1, ..., v_{T+1}^K$

1: residual $h_{T+1}^* = h_{T+1}$
2: $\eta = \frac{T-l}{T+1} \cdot (1 - \epsilon)$
3: **for** $k$ in $range(n)$ **do**
4:     update $v_{T+1}^k$ using $h_{T+1}^*$ with equation (20)
5:     update $h_{T+1}^*$ with equation (19)
6: **end for**
7: **if** $n < K_{\max}$ and $\frac{\|h_{T+1}^*\|}{\|h_{T+1}\|} > \delta$ **then**
8:     add a new component $v_{T+1}^{n+1} = h_{T+1}^*$
9:     $n = n + 1$
10: **end if**

---

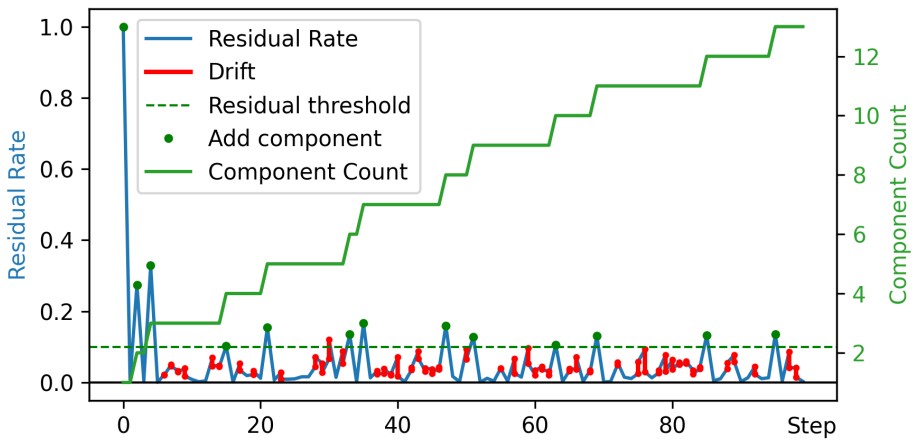

Figure 4: The update of principal components. If the residual rate is over the threshold, we add a new component for it. The red line shows that we track the drift by adjusting the tracking factor $\epsilon$, whose increase mostly reduces the residual rate for better drift tracking.

drift of the average gradient direction $\mathbf{cos}\langle G_T, \bar{G}_T \rangle$ and LoRA B vectors $\frac{1}{r} \sum_{n=1}^r \mathbf{cos}\langle \tilde{\beta}_n, \beta_n \rangle$ in the same way described under Figure 2. We test all data points (batch) of each CL Benchmark dataset; the results are shown in Figure 5.

## C   ADDTIONAL EXPERIMENTAL DETAILS

For the Online PCA in our method, we fix the amnestic factor $l = 2$, following the recommendation of Weng et al. (2003). The tracking factor is that $\epsilon \in (0, \epsilon_{\max})$, and the empirical value is $\epsilon_{\max} = 0.1$. The tracking factor is that $\epsilon$ is adjusted with the increase and decrease of the residual rate, and the residual threshold $\delta = 0.1$ for adding components automatically. For each new task, we enhance the maximum principal component number $K_{\max}$ by 48. Apart from the hyperparameter study in Table 5, we also investigated the combination of different tracking factors $\epsilon_{\max}$ and residual thresholds $\delta$; the results are shown in Table 7. Note that a higher tracking factor $\epsilon_{\max}$ and a lower residual threshold $\delta$ mean stronger tracking, as the former makes a stronger update of principal components, and the latter makes adding new components more frequent. Our method (**DOC**) takes $\epsilon_{\max} = 0.1, \delta = 0.1$; we refer to the one with the stongest tracking ($\epsilon_{\max} = 0.2, \delta = 0.05$) as **DOC-strong** and the weakest one ($\epsilon_{\max} = 0.05, \delta = 0.2$) as **DOC-weak** in Table 1 and Table 3.

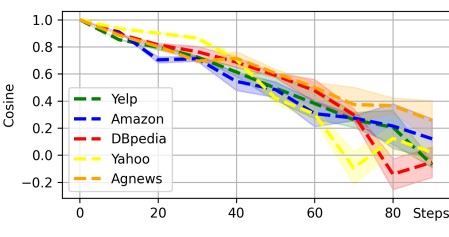 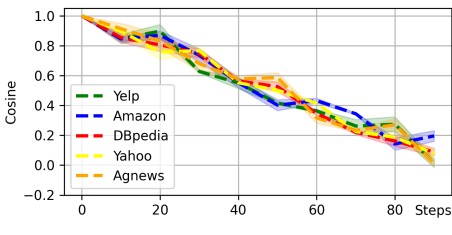

(a) Gradient with average (b) LoRA B

Figure 5: Extended study of Figure 2 (a). We further demonstrate the drift of functional directions on each dataset in the CL Benchmark using LLaMA-7B. Figure (a) shows the drift of the average gradient direction, *i.e.* $\cos\langle G_T, \bar{G}_T \rangle$; Figure (b) shows the drift of LoRA $B$ vectors, *i.e.* $\frac{1}{r} \sum_{n=1}^{r} \cos\langle \tilde{\beta}_n, \beta_n \rangle$. Computational details are shown under Figure 2. The results are the average of data points in each dataset, with standard deviation shown. The dropping similarity between functional directions and their historical values further verifies the drift of functional directions.

Table 7: Average Accuracy (AA) results of standard CL Benchmark on LLaMA-7B with different tracking factor $\epsilon_{\max}$ and residual threshold $\delta$. The results are the average of task orders 1-3.

|  | $\epsilon_{\max} = 0.05$ | $\epsilon_{\max} = 0.1$ | $\epsilon_{\max} = 0.2$ |
|---|---|---|---|
| $\delta = 0.05$ | 76.4 | 77.4 | 68.6 |
| $\delta = 0.1$ | 73.7 | 77.7 | 76.6 |
| $\delta = 0.2$ | 64.2 | 76.9 | 75.2 |

We follow O-LoRA(Wang et al., 2023a) and Progressive Prompt (Razdaibiedina et al., 2023) for the following continual learning settings:

**Dataset details**. Table 8 shows details of the datasets we employ for continual learning experiments, along with their evaluation metrics. Overall, we used datasets from CL benchmark (Zhang et al., 2015), GLUE (Wang et al., 2018), and SuperGLUE (Wang et al., 2019) benchmarks, adding the IMDB movie reviews (Maas et al., 2011). We randomly sample 100-10000 samples for each dataset, depending on their size, and fine-tune for 1000 steps for each incoming dataset in streaming data.

**Task sequence of continual learning**. The task orders used for our CL experiments across LLaMA and T5 models are shown in Table 9.

**Prompts for different tasks**. Table 10 shows prompts for different tasks. NLI denotes natural language inference, including MNLI, RTE, CB. SC denotes sentiment analysis, including Amazon, Yelp, SST-2, IMDB. TC denotes topic classification, including AG News, DBpedia, Yahoo.

## D  DETAILS FOR COMPUTATION

**Extract input vector $x$ with token average**. In our method, we extract LoRA increment $\mathbf{d}Wx$ as the functional direction, where $x$ is the input *vector* of the module with the parameter matrix $W$. Note that in a transformer model, the input, say $X$, to $W$ is several vectors, that is:

$$X = (x_1, x_2, ..., x_n) \tag{21}$$

where $N$ is the number of input tokens, $x_n$ is the input vector at the place of the $n$-th token. Common methods to represent inputs $x_1, x_2, ..., x_n$ with a single vector $x$ include computing their average or taking the last vector. For stability of computation, we employ the average method, that is:

$$x = \frac{1}{N} \sum_{n=1}^{N} x_n \tag{22}$$

**Standarization of LoRA increment $h$ for PCA**. We employ the LoRA increment $h$(computed with equation (5)) as the functional direction in our method. As there is no scale difference in gradients, we omit normalization, following (Cardot & Degras, 2018). Note that we are concerned with only the directions of $h$, so we also conduct no centralization for $h$ at the beginning. Note that in this

Table 8: The details of 15 datasets used in the CL experiments, following O-LoRA and Progressive Prompt. NLI denotes natural language inference, QA denotes the question and answer task.

| Dataset name | Category | Task | Domain | Metric |
|---|---|---|---|---|
| 1. Yelp | CL Benchmark | sentiment analysis | Yelp reviews | accuracy |
| 2. Amazon | CL Benchmark | sentiment analysis | Amazon reviews | accuracy |
| 3. DBpedia | CL Benchmark | topic classification | Wikipedia | accuracy |
| 4. Yahoo | CL Benchmark | topic classification | Yahoo Q&A | accuracy |
| 5. AG News | CL Benchmark | topic classification | news | accuracy |
| 6. MNLI | GLUE | NLI | various | accuracy |
| 7. QQP | GLUE | paragraph detection | Quora | accuracy |
| 8. RTE | GLUE | NLI | news, Wikipedia | accuracy |
| 9. SST-2 | GLUE | sentiment analysis | movie reviews | accuracy |
| 10. WiC | SuperGLUE | word sense disambiguation | lexical databases | accuracy |
| 11. CB | SuperGLUE | NLI | various | accuracy |
| 12. COPA | SuperGLUE | QA | blogs, encyclopedia | accuracy |
| 13. BoolQA | SuperGLUE | boolean QA | Wikipedia | accuracy |
| 14. MultiRC | SuperGLUE | QA | various | accuracy |
| 15. IMDB | SuperGLUE | sentiment analysis | movie reviews | accuracy |

Table 9: Different orders of task sequences used for continual learning experiments. Orders 1-3 correspond to the standard CL benchmark, orders 4-6 are long chain of tasks, following O-LoRA and Progressive Prompt.

| Order | Task Sequence |
|---|---|
| 1 | dbpedia → amazon → yahoo → ag |
| 2 | dbpedia → amazon → ag → yahoo |
| 3 | yahoo → amazon → ag → dbpedia |
| 4 | mnli → cb → wic → copa → qqp → boolqa → rte → imdb → yelp → amazon → sst-2 → dbpedia → ag → multirc → yahoo |
| 5 | multirc → boolqa → wic → mnli → cb → copa → qqp → rte → imdb → sst-2 → dbpedia → ag → yelp → amazon → yahoo |
| 6 | yelp → amazon → mnli → cb → copa → qqp → rte → imdb → sst-2 → dbpedia → ag → yahoo → multirc → boolqa → wic |

case, the first few principal component represents the weighted historical average (17), and the residual cut in equation (19) will deduct the average and thus reach certain centralization. Also, the effect of other normalization methods designed for LoRA increments or gradients deserves further investigation.

# E  ADDITIONAL RELATED WORKS

The following methods have been developed for LLM continual learning. They can be categorized into the following types: Rehearsal-based , Architecture-based , Prompt-based, and Regularization-based approaches. A brief summary is in Table 12.

**Rehearsal-based approach** (de Masson D'Autume et al., 2019; Mok et al., 2023; Huang et al., 2021) try to remind the model of historical tasks and thus avoid forgetting. However, there are growing restoration costs as tasks accumulate, and privacy issues in gaining historical training data.

**Architecture-based approach** (Jang et al., 2023; Wang et al., 2024; Peng et al., 2024) train multiple expert models for each task. However, when it comes to unseen tasks, there is no proper expert to use, which destroys the generalization ability of models.

**Prompt-based approach** L2P(Wang et al., 2022), LFPT5(Qin & Joty, 2022), and Progressive Prompts(Razdaibiedina et al., 2023) add prompts during the inference of the model. This approach

Table 10: Instructions for different tasks, following O-LoRA and Progressive Prompt.

| Task | Prompts |
|------|---------|
| NLI | What is the logical relationship between the "sentence 1" and the "sentence 2"? Choose one from the options. |
| QQP | Whether the "first sentence" and the "second sentence" have the same meaning? Choose one from the options. |
| SC | What is the sentiment of the following paragraph? Choose one from the options. |
| TC | What is the topic of the following paragraph? Choose one from the options. |
| BoolQA | According to the following passage, is the question true or false? Choose one from the options. |
| MultiRC | According to the following passage and question, is the candidate answer true or false? Choose one from the options. |
| WiC | Given a word and two sentences, whether the word is used with the same sense in both sentences? Choose one from the options. |

is lightweight, but when the fine-tuning information gets large, the prompt will not be able to cover it.

**Regularization-based Approach** EWC(Kirkpatrick et al., 2017; Zenke et al., 2017), LwF(Li & Hoiem, 2017), OGD(Farajtabar et al., 2020), and O-LoRA(Wang et al., 2023a) limit the update of model parameters to preserve the historical ability of the model. Their edge is that no historical data or extra architecture is required. We lay emphasis on the orthogonal methods, including OGD and O-LoRA, as we employ orthogonal cuts to avoid changing historical parameter settings and preserve historical functions.

**Orthogonal methods** The key point of orthogonal methods is to avoid wrecking the parameter subspace of historical tasks when fine-tuning on the latest task, and the method is to make the parameter space of new tasks orthogonal to the historical ones. Representative methods, including Orthogonal Gradient Descent (OGD)(Farajtabar et al., 2020) and Orthogonal Subspace Learning(O-LoRA)(Wang et al., 2023a), have been proven effective in preventing catastrophic forgetting. OGD forces the gradient descent to be orthogonal to the gradient directions of historical tasks. That is:

$$G_T \perp G_t \quad t = 1, 2, ..., T - 1 \tag{23}$$

where $G_t = \nabla_\theta L_t$ is the gradient direction of the $t$ th task. O-LoRA tries to make the LoRA $B$ matrix in equation (2) orthogonal to that of historical LoRA modules. That is:

$$\beta_T^i \perp \beta_t^j \quad t = 1, 2, ...T - 1 \quad i, j = 1, 2, ..., r \tag{24}$$

where $\beta_t^i$ is the $i$ th colomun vector of $B_{m \times r}$ matrix fine-tuned in the $t$ th task, that is $B = (\beta_1, \beta_2, ..., \beta_r)$.

# F    LIMITATIONS

While the proposed method has an outstanding performance in empirical evaluations, we discuss its potential limitations as follows.

**Scalability**. In more complex scenarios with a large number of tasks, such as hundreds of tasks, the principal component pool expands as we add new components during the fine-tuning process, imposing a growing load for computation. The empirical scale of the expansion is approximately 40 components for each task, as shown in Figure 6. The size of these components is approximately 15MB and is acceptable in the settings of our experiments. However, in the case of hundreds of tasks, the performance and applicability of our method require further investigation.

**Task identification**. Although our method requires no task identification during inference, it is still required during the continual fine-tuning process. Exploring methods for task-agnostic training would be a valuable future direction. This is further discussed in Future Directions.

Table 11: The accuracy on the MMLU benchmark of LLaMA-7B before and after continual learning (CL) on the CL benchmark. The results are the average of task orders 1-3. Note that with MMLU being a four-classification problem, a 25% accuracy equates to random guessing.

| | MMLU Accuracy |
|---|---|
| Original model | 32.3 |
| Alpaca LoRA fine-tuned model | 36.0 |
| Seq LoRA CL after Alpaca LoRA | 26.2 |
| O-LoRA CL after Alpaca LoRA | 30.1 |
| DOC CL after Alpaca LoRA | 29.4 |
| O-LoRA throughout Alpaca and CL | 32.1 |
| **DOC throughout Alpaca and CL** | **34.6** |

Table 12: The comparison of continual learning methods. Specifically, **RF** indicates whether the method is rehearsal-free. **TIF** indicates whether the task ID is free during inference. Compared to regularization-based methods, other methods have extra settings or computational overheads.

| | | RF | TIF | Inference costs |
|---|---|---|---|---|
| Rehearsal-based | MBPA++ (de Masson D'Autume et al., 2019) | | ✓ | |
| | IDBR (Huang et al., 2021) | | ✓ | |
| Architecture-based | EIP (Wang et al., 2023b) | ✓ | ✓ | |
| | SLM (Peng et al., 2024) | ✓ | ✓ | Expert selection |
| | Expert LMs (Jang et al., 2023) | | ✓ | |
| | MoCL (Wang et al., 2024) | ✓ | ✓ | |
| Prompt-based | L2P (Wang et al., 2022) | ✓ | ✓ | |
| | LFPT5 (Qin & Joty, 2022) | | ✓ | Additional prompts |
| | ProgPrompt (Razdaibiedina et al., 2023) | ✓ | | |
| Regularizatoin-based | EWC (Kirkpatrick et al., 2017) | ✓ | ✓ | |
| | LwF (Li & Hoiem, 2017) | ✓ | | |
| | OGD (Farajtabar et al., 2020) | ✓ | ✓ | |
| | O-LoRA (Wang et al., 2023a) | ✓ | ✓ | |
| | **DOC**(ours) | ✓ | ✓ | |

**Generalization Ability**. As our method is targeted at preserving historical functions, it has no recognition of unseen tasks. Its generalization ability deserves further investigation. We propose the following empirical demonstration of the impact of our method on the generalization ability of the model. Following O-LoRA (Wang et al., 2023a), we start with a fine-tuned LLaMA-7B language model on the Alpaca (Taori et al., 2023) dataset. After conducting continual learning on the CL benchmark (Zhang et al., 2015), we test the model on the MMLU benchmark (Hendrycks et al., 2021b;a), composed of unseen tasks. The results are shown in Table 11. Compared to the original model, SeqLoRA and our method (DOC) suffer from forgetting (accuracy respectively drops from 36.0 to 26.2 and 29.4). This is because of the lack of information about unseen tasks during continual learning. In the experimental settings, the issues above limit the practicality of DOC.

Note that we fine-tune the model on Alpaca at the beginning, so that continual learning also triggers the forgetting of Alpaca. What if we mitigate this forgetting with CL methods? We further investigate the effect of continual fine-tuning the model on the Alpaca and CL benchmark with CL methods applied **throughout from start to end**, which makes the Alpaca **visible** to the methods. Note that MMLU is still unseen during the continual fine-tuning process in this setting. The results show the enhanced performance (an accuracy of 34.6 for DOC and 32.1 for O-LoRA, compared to 29.4 and 30.1 in the former experiment where Alpaca is invisible to the methods). An explanation is that the methods avoid forgetting Alpaca, which is a general dataset that assists in the initialization of crucial functional directions of the model, thus aiding in the preservation of crucial functions for unseen tasks. The results also indicate that the generalization of our method, which preserves the ability on unseen tasks with a general dataset, is better compared to O-LoRA. It inspires the practical deployment of DOC to initialize on a general dataset beforehand.

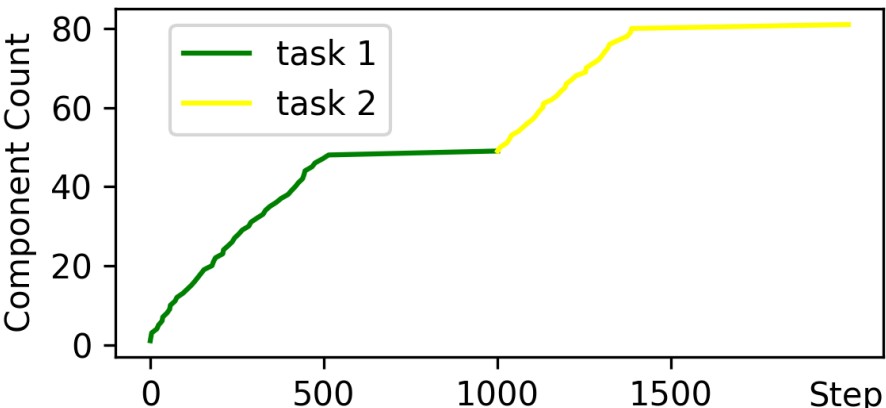

Figure 6: The expansion of principal components in the first 2 tasks. Note that we do not limit the maximum principal component number in this experiment, *i.e.*, $K_{max} = +\infty$. As the number of principal components increases, it reaches a point where no extra component is required, indicating that the current components are adequate to cover a large enough part of the LoRA increment.

## G    FUTURE DIRECTOINS

**The interpretability of principal components**. We employ PCA in our method for functional direction tracking. Another edge of PCA is that the components extracted are statistically independent of each other; thus, each component represents an individual unit, as proposed by (Michaud et al., 2023). The individuality of these components provides chances for model deconstruction and better interpretability, and it is possible to find the exact meaning of each component, *e.g.*, semantic function, logic function, certain knowledge, *etc.*, through empirical methods. It is a promising direction for interpretable learning based on model deconstruction.

**Automated task ID recognition** As mentioned before, exploring methods for task-agnostic training would be valuable. It deserves further investigation into the characteristics of the principal components extracted from a specific task, which assists in the distinction of different tasks.

