# OpenReview forum: "Dynamic Orthogonal Continual Fine-tuning for Mitigating Catastrophic Forgetting of LLMs"
_ICLR.cc/2026/Conference — Submitted to ICLR 2026_

### Official Review · Reviewer_3uqt · 2025-10-29

**Soundness:** 3
**Presentation:** 2
**Contribution:** 3
**Rating:** 4
**Confidence:** 4

**Summary:**

This paper introduces Dynamic Orthogonal Continual (DOC) fine-tuning for LLM continual learning. The authors identify drift of functional directions during fine-tuning as a key reason regularization-based methods fail. They then propose tracking this drift with a modified Online Principal Component Analysis (PCA) and orthogonally cutting new gradients with respect to the tracked historical functional directions. Experiments show that the method mitigates catastrophic forgetting and outperforms prior methods.

**Strengths:**

1. The paper clearly identifies drift in functional directions during fine-tuning as the
key issue for regularization-based approaches, and proposes an intuitive method that
tracks and updates these directions using gradients and LoRA increments, then applies
an orthogonal cut.
2. By using modified online PCA, the method updates principal components from the most recent
data without storing historical data. The authors justify adopting Candid Covariance-free
Incremental PCA because it allows components to be added freely and has lower
computational overhead, demonstrating efficient tracking with a modified online PCA.
3. On LLaMA-7B and LLaMA-13B and on T5-Large, the paper reports Average Accuracy,
Backward Transfer, and Forward Transfer on standard continual learning benchmarks and
on long task chains, and includes DOC ablations to highlight the impact of tracking.

**Weaknesses:**

1. The rationale for choosing online PCA is largely descriptive, and there is no comparative study among alternative online PCA methods. Although the paper lists approaches such as incremental PCA and stochastic approximation methods and gives reasons for preferring CCIPCA, it does not provide head-to-head comparisons under the same continual learning setup, including tracking quality, compute, and memory.
2. Comparisons center on EWC, LwF, and O-LoRA; other-based methods are excluded as not comparable due to additional settings, and oracle methods are only upper bounds. This narrow and dated selection suggests that more recent baselines are missing, making the coverage of related work and empirical comparisons limited.
3. Motivation leans too heavily on Figure 2, which reports cosine similarity for a particular datapoint x, y and an average over randomly chosen datapoints with standard deviation. It stops short of generalizing the drift analysis beyond that setup.

**Questions:**

1. Please include comparative experiments within the same continual learning setup that evaluate the multiple online PCA approaches described in the paper against the modified CCIPCA, reporting tracking quality, Average Accuracy, Backward Transfer, Forward Transfer, and computational costs.
2. Beyond a single datapoint and the average over randomly chosen datapoints in Figure 2, please present drift analyses under more diverse conditions directly in the motivation.
3. Please provide additional experiments on the tracking hyperparameters, including the amnesic factor, the tracking factor, and the residual threshold, and present residual rate and coverage trends in greater detail.
4. In addition to the setting that freezes updates of the principal components, please evaluate how varying the tracking strength affects Average Accuracy, Backward Transfer, and Forward Transfer.

---

> ### Author Response · Authors · 2025-11-21
> **Response to Reviewer 3uqt (Part 1/3)**
>
> Dear Reviewer 3uqt:
>
> Thank you for your valuable feedback and for recognizing our work for identifying functional drift, efficient tracking via modified online PCA, and comprehensive experiments with ablations. In response to your concerns, we have carefully revised the manuscript and provide detailed responses below.
>
> ---
>
> **Q1 & W1**: Please include comparative experiments within the same continual learning setup that evaluate the multiple online PCA approaches described in the paper against the modified CCIPCA, reporting tracking quality, Average Accuracy, Backward Transfer, Forward Transfer, and computational costs.
>
> The rationale for choosing online PCA is largely descriptive, and there is no comparative study among alternative online PCA methods. Although the paper lists approaches such as incremental PCA and stochastic approximation methods and gives reasons for preferring CCIPCA, it does not provide head-to-head comparisons under the same continual learning setup, including tracking quality, compute, and memory.
>
> **A1**: Thanks for providing a noteworthy aspect. We further discuss the effect of using different Online PCA techniques in our method in Section 4.3, including IPCA, SGA, GHA, and SNL techniques (denoted as DOC w. IPCA/SGA/GHA/SNL in the tables).  Their tracking qualities are shown in Figure 3 (a), average Accuracies are shown in Table 1, BWT, FWT scores are in Table 3, and computational costs are in Table 6. Part of the results are shown below.
>
> Results show that while the continual learning performances of different Online PCA techniques are similar, other online PCA methods introduce significantly more computational overhead. As shown in Table 7, DOC with other online PCA methods requires at least 40% (0.46s→0.66s) more computational cost than DOC (with CCIPCA), validating its superiority on updating efficiency. The reason behind this difference is that other online PCAs have to update all the principal components from the beginning, which does not suit a real-world continual learning scenario and adds extra computational loads. Therefore, the **CCIPCA is an optimal choice due to its capacity to freely add new components and its light computational overhead**.
>
> *Part of Table 1 (from Section 4.2): Average Accuracy (AA) of different continual methods on LLaMA-7B, including DOC with (w.) different online learning techniques.*
>
> | Method | Order 1 | Order 2 | Order 3 | Average (↑) | Order 4 | Order 5 | Order 6 | Average (↑) |
> | --- | --- | --- | --- | --- | --- | --- | --- | --- |
> | DOC w. IPCA | 77.6 | 79.4 | 74.7 | 77.2 | 69.7 | 72.6 | 73.8 | 72.0 |
> | **DOC w. SGA** | 78.2 | **79.4** | **75.2** | 77.6 | 70.4 | 73.0 | 69.9 | 71.1 |
> | **DOC w. GHA** | 79.3 | 77.2 | 74.9 | 77.1 | 69.8 | 70.5 | 68.4 | 69.6 |
> | **DOC w. SNL** | 79.8 | 78.5 | 74.5 | 77.6 | 71.1 | 72.1 | 68.5 | 70.5 |
>
> ---
>
> *Part of Table 3 (from Section 4.2): Average BWT and FWT scores of different continual methods on LLaMA-7B, including DOC with (w.) different Online PCA techniques.*
>
> |  |      Standard CL | Benchmark |         Long chain | of tasks |
> | --- | --- | --- | --- | --- |
> | Method | BWT(↑) | FWT(↑) | BWT | FWT |
> | **DOC w. IPCA** | -0.8 | 1.2 | -4.8 | -0.3 |
> | **DOC w. SGA** | -0.7 | **1.6** | -5.6 | -0.5 |
> | **DOC w. GHA** | -1.0 | 1.2 | -7.9 | **0.2** |
> | **DOC w. SNL** | -0.7 | 1.6 | -6.0 | -0.7 |
>
> *Table 6 (from Section 4.3): Average clock time per fine-tuning step (update over one batch) of DOC with (w.) different Online PCA techniques. The model is LLaMA-7B.*
>
> | Method | DOC | DOC w. IPCA | DOC w. SGA | DOC w. GHA | DOC w. SNL |
> | --- | --- | --- | --- | --- | --- |
> | Clock Time | 0.46s | 0.92s | 0.88s | 0.67s | 0.66s |

---

> > ### Author Response · Authors · 2025-11-21
> > **Response to Reviewer 3uqt (Part 2/3)**
> >
> > **Q2 & W3**: Beyond a single datapoint and the average over randomly chosen datapoints in Figure 2, please present drift analyses under more diverse conditions directly in the motivation.
> >
> > Motivation leans too heavily on Figure 2, which reports cosine similarity for a particular datapoint x, y and an average over randomly chosen datapoints with standard deviation. It stops short of generalizing the drift analysis beyond that setup.
> >
> > **A2**: Thanks for your kind suggestion.  We present drift analyses under all the CL Benchmark datasets in Figure 5 in Appendix B, where the dropping similarity between functional directions and their historical values further verifies the drift of functional directions. This observation is consistent with our datapoint-level experiments (Figure 2) and further strengthens our motivation.
> >
> > **Q4**: In addition to the setting that freezes updates of the principal components, please evaluate how varying the tracking strength affects Average Accuracy, Backward Transfer, and Forward Transfer.
> >
> > **A4**: You denote a point that is worth trying. We change the tracking factor and the residual threshold to vary the strength of tracking, and introduce DOC-strong and DOC-weak. Here, ‘strong’ and ‘weak’ refer to tracking strengths, as these two hyperparameters affect the tracking strengths (a higher tracking factor and a lower residual threshold mean stronger tracking, as the former makes a stronger update of principal components, and the latter makes adding new components more frequent). Their performances on continual learning are shown in Table 1 and Table 3. Since a moderate tracking strength makes an optimal tracking quality, its continual learning performance is better, consequently.
> >
> > *Part of Table 1 (from Section 4.2): Average Accuracy (AA) of different continual methods on LLaMA-7B*
> >
> > | Method | Order 1 | Order 2 | Order 3 | Average (↑) | Order 4 | Order 5 | Order 6 | Average (↑) |
> > | --- | --- | --- | --- | --- | --- | --- | --- | --- |
> > | **DOC-strong** | 78.0 | 75.7 | 75.1 | 76.2 | 70.0 | 67.7 | 68.3 | 68.6 |
> > | **DOC-weak** | 70.2 | 69.5 | 71.6 | 70.4 | 63.4 | 64.6 | 64.6 | 64.2 |
> >
> > ---
> >
> > *Part of Table 3 (from Section 4.2): Average BWT and FWT scores of different continual methods on LLaMA-7B*
> >
> > | Method | Standard CL | Benchmark | Long chain | of tasks |
> > | --- | --- | --- | --- | --- |
> > |  | BWT(↑) | FWT(↑) | BWT | FWT |
> > | **DOC-strong** | -1.9 | 1.0 | -7.7 | -1.1 |
> > | **DOC-weak** | -7.8 | -0.3 | -13.6 | 0.1 |

---

> ### Author Response · Authors · 2025-11-21
> **Response to Reviewer 3uqt (Part 3/3)**
>
> **Q3**: Please provide additional experiments on the tracking hyperparameters, including the amnesic factor, the tracking factor, and the residual threshold, and present residual rate and coverage trends in greater detail.
>
> **A3**: We appreciate your careful concern. We present studies on different combinations of the tracking factor and the residual threshold in Section 4.3 ablation study, and Appendix C.  The continual learning accuracies of different combinations of these two hyperparameters are in Table 7. We also present the tracking quality under different combinations of these two hyperparameters in Figure 3 (b), using DOC-strong and DOC-weak with different tracking factors and residual thresholds, as mentioned above in A4.
>
> The result of tracking quality (shown in Figure 3 (b)) shows that a moderate tracking strength is more stable compared to a strong one and more effective compared to a weak one. Its continual learning performance is better (shown in Table 7), consequently.
>
> *Table 7 (from Appendix C): Average Accuracy (AA) results of standard CL Benchmark on LLaMA-7B with different tracking factor $\epsilon_{\max}$ and residual threshold $\delta$. The results are the average of task orders 1-3.*
>
> |  | ε_max = 0.05 | ε_max = 0.1 | ε_max = 0.2 |
> | --- | --- | --- | --- |
> | δ = 0.05 | 76.4 | 77.4 | 68.6 |
> | δ = 0.1 | 73.7 | **77.7** | 76.6 |
> | δ = 0.2 | 64.2 | 76.9 | 75.2 |
>
> The residual rate and coverage trends are used for the algorithm decision and are controlled by the residual threshold, i.e., the algorithm keeps the residual rate under the residual threshold to ensure a good coverage. Specifically, if the residual rate rises, it means that we capture a functional direction drift, and enhance the tracking factor for better tracking, and vice versa if the residual rate falls. The residual rate is also used to add new principal components. If the residual rate is over the residual threshold, it means that a new functional direction is captured, and we add a new principal component to restore this functional direction. More details on their usage in our method are provided in Appendix A.
>
> You also denoted the amnesic factor; here we kindly note that the amnesic factor is fixed ($l=2$) following the recommendation of Weng et al. (2003), so we do not change that.
>
> ---
>
> **W2**: Comparisons center on EWC, LwF, and O-LoRA; other-based methods are excluded as not comparable due to additional settings, and oracle methods are only upper bounds. This narrow and dated selection suggests that more recent baselines are missing, making the coverage of related work and empirical comparisons limited.
>
> **A5**: We respectfully argue that other baselines are **not comparable due to different settings** (as you acknowledged). Moreover, please note that O-LoRA is the SOTA regularization-based method, and we also include Progressive Prompt, the SOTA other-based method, in our experiments. This makes sure that the comparisons in our experiments are grounded and valid.
>
> We truly appreciate your valuable and detailed feedback. If you have any further questions or concerns, please let us know.

---

> > ### Author Response · Authors · 2025-11-27
> >
> > Dear Reviewer 3uqt,
> >
> > It is our great honor to have you as a reviewer for our paper. We sincerely appreciate the time and effort you have devoted to your review. We have carefully clarified or addressed all the concerns you raised, and sincerely hope that these clarifications and revisions have resolved them.
> >
> > As the discussion phase is nearing its end, we just wanted to gently check whether you have any remaining concerns. We would be very happy to clarify anything further if needed.
> >
> > Thank you again for your thoughtful feedback. Wishing you a lovely day!
> >
> > Warmly,
> >
> > Authors

---

### Official Review · Reviewer_rKEH · 2025-10-31

**Soundness:** 3
**Presentation:** 3
**Contribution:** 3
**Rating:** 6
**Confidence:** 4

**Summary:**

This work addresses the continual fine-tuning of large language models (LLMs). The authors first identified the drift of functional directions as the main cause of the failure in regularization-based CL for LLMs. Based on this observation, they proposed Dynamic Orthogonal Continual (DOC) to address this issue. DOC consists of two components. First, the authors track the increment of LoRA as the effective functional direction. An online PCA is applied to dynamically update the stored functional directions in sequential learning. Second, the authors applied an orthogonal projection to avoid knowledge interference. Extensive experiments have demonstrated the effectiveness of the proposed method.

**Strengths:**

1. The paper is well motivated

The proposed method is derived from an observation and has successfully alleviated the issue.

2. The presentation is good

Many illustrative figures and texts are useful to help understand the phenomenon and the context

3. The proposed method is effective

**Weaknesses:**

1. Some details of the method are not clear enough

2. Presentation of Figure.1 can be improved

3. Applicability to other parameter-efficient-tuning CL methods

Please see the details below in the Question section.

**Questions:**

I would love to have a bit more details about the implementation details

1. Is the LoRA shared across CL tasks or different LoRA adapters are used for each task?

It is not clearly stated in the paper.

2. How to initialize the set of h to calculate PCA?

Is the initial set of h calculated by the end of task?

3. How to compute cosine similarity if the vectors do not match?

According to line 179 and lines 251-259, it seems the coordinate vector is dependant on the principle components. It is likely the coord(h1) and coord(h5) do not have the same number of principle components. If this is the case, how to compare these two vectors to calculate the cosine similarity?

4. Presentation of Figure.1 can be improved

The two sub-figures here are hard to understand.

5. Applicability to other parameter-efficient-tuning CL methods

I am wondering if the proposed method is compatible with CL methods based on other parameter-effecient tuning techniques, such as prompt tuning.

**Details Of Ethics Concerns:**

No concern.

---

> ### Author Response · Authors · 2025-11-21
> **Response to Reviewer rKEH**
>
> Dear Reviewer rKEH:
>
> Thank you for your insightful feedback and for recognizing our work for the motivation of functional drift, the presentation of a novel method, and experiments showing method effectiveness. In response to your concerns, we have carefully revised the manuscript and provide detailed responses below.
>
> ---
>
> **W1 & Q1** : Some details of the method are not clear enough; Is the LoRA shared across CL tasks, or different LoRA adapters are used for each task? It is not clearly stated in the paper.
>
> **A1**: We appreciate your rigorous approach to technical details. The LoRA is shared across CL tasks, and we do not introduce new LoRA modules for new tasks in our method (Summarized in Algorithm 1). We also take your advice and make further clarification in Section 2.2.
>
> ---
>
> **Q2**: How to initialize the set of h to calculate PCA? Is the initial set of h calculated by the end of task?
>
> **A2**: Thanks for your question, but we have to respectfully note your misunderstanding on Online PCA.  Actually, the set of h is not restored. As described in Section 3.2 and Appendix A, our PCA method updates the principal components using only one incoming LoRA increment (h), so there is no need to restore the set of historical h.
>
> Maybe your question is ‘how to initialize the set of principal component (v) in the experiment of Figure 2 (b)’? In Figure 2 (b), we initialize principal components (say {v_1,…, v_n}) during the first task (say task A) using our tracking method that adds and updates the principal components that restore the functional directions; the plot is for the next task (say task B). Though we still add new components in task B, we only take {v_1, …, v_n} into consideration when comparing coordinate vectors. Note that {v_1, …, v_n} are continually updated during task B, and they contain the historical functional directions in task A, so the plot demonstrates the effectiveness of tracking.
>
> ---
>
> **Q3**: How to compute cosine similarity if the vectors do not match?
>
> According to line 179 and lines 251-259, it seems the coordinate vector is dependant on the principle components. It is likely the coord(h1) and coord(h5) do not have the same number of principal components. If this is the case, how to compare these two vectors to calculate the cosine similarity?
>
> **A3**: We appreciate your sharp observation. Your concern is that the vectors with different numbers of principal components as their coordinate bases have different numbers of dimensions, but this problem is avoided in our experiment in Figure 2, where we only take the principal components (the set of v) initialized beforehand when computing coordinate vectors. Specifically, in Figure 2 (b), we initialize principal components (say {v_1,…, v_n}) during the first task (say task A), and the plot is for the next task (say task B). Though we still add new components in task B, we only take {v_1, …, v_n} into consideration when comparing coordinate vectors. Note that {v_1, …, v_n} are continually updated during task B, and they contain the historical functional directions in task A, so the plot demonstrates the effectiveness of tracking.
>
> ---
>
> **Q4**: Presentation of Figure 1 can be improved. The two sub-figures here are hard to understand.
>
> **A4**: Thanks for your kind advice. We add denotations under the figures to describe the correlation of each part of the figures to our algorithm, ensuring a better correspondence to our writings. This may help understand the figures
>
> ---
>
> **Q5**: Applicability to other parameter-efficient-tuning CL methods
>
> I am wondering if the proposed method is compatible with CL methods based on other parameter-efficient tuning techniques, such as prompt tuning.
>
> **A5**: Thanks for your insightful perspective. We clarify that our method is specifically developed for LoRA continual fine-tuning, but extension to other parameter-effecient tuning can be a valuable future direction. We also have to kindly note that prompt tuning is conducted in the prompt space rather than in a parameter-efficient tuning circumstance.
>
> ---
>
> We truly appreciate your valuable and detailed feedback. If you have any further questions or concerns, please let us know.

---

### Official Review · Reviewer_K2j5 · 2025-11-01

**Soundness:** 2
**Presentation:** 3
**Contribution:** 2
**Rating:** 2
**Confidence:** 3

**Summary:**

This paper addresses catastrophic forgetting in continual learning for large language models (LLMs). The authors identify the drift of functional directions during fine-tuning as a key factor behind the failure of existing regularization-based methods in long-term continual learning. To mitigate this issue, they propose Dynamic Orthogonal Continual (DOC) fine-tuning, which dynamically tracks and updates these functional directions while enforcing gradient orthogonality to reduce interference between tasks. Experiments on multiple LLM continual learning benchmarks show that the proposed method effectively alleviates forgetting and outperforms regularization-based approaches.

**Strengths:**

1. The idea is clear and intuitive. This paper considers tracking the drift of ffunctional directions that regularization-based method ignored and dynamically updates them during the fine-tuning process.
2. This paper is well-written and easy to follow.

**Weaknesses:**

1. The statement “We reveal the drift of function directions in the fine-tuning process, which explains why regularization-based approaches fail in long-term LLM continual learning” is somewhat overstated. The described phenomenon of drifting functional directions appears more as an empirical observation rather than a definitive causal explanation. From a theoretical standpoint, one fundamental reason for forgetting in neural networks is the non-convex nature of their optimization landscape. When the model is fine-tuned on new data distributions, the optimizer converges to a different local minimum, shifting the basin of attraction and consequently degrading previously learned representations.
2. Since the paper’s core claim is that dynamically updated regularization mitigates forgetting, the experiments should explicitly probe (1) task-order sensitivity (change task orders) and (2) sequence-length robustness (Vary the number of tasks ).These ablations would substantiate that the proposed dynamic regularization is not brittle to ordering effects and remains effective over long horizons.

**Questions:**

see above

---

> ### Author Response · Authors · 2025-11-21
> **Response to Reviewer K2j5**
>
> Dear Reviewer  K2j5:
>
> We appreciate your acknowledgment of our work on tracking the drift of functional directions that prior regularization-based methods ignored, dynamically updating them during the fine-tuning process. In response to your concerns, we have carefully revised the manuscript and provide detailed responses below.
>
> ---
>
> **Q1**: The statement “We reveal the drift of function directions in the fine-tuning process, which explains why regularization-based approaches fail in long-term LLM continual learning” is somewhat overstated. The described phenomenon of drifting functional directions appears more as an empirical observation rather than a definitive causal explanation. From a theoretical standpoint, one fundamental reason for forgetting in neural networks is the non-convex nature of their optimization landscape. When the model is fine-tuned on new data distributions, the optimizer converges to a different local minimum, shifting the basin of attraction and consequently degrading previously learned representations.
>
> **A1**: We appreciate your insightful perspective. The shifting basin in the loss landscape between different tasks is an enlightening point of view in understanding continual learning problems, but it does not conflict with our grounded observation of functional direction drift, as the former is a global view on the whole loss landscape, while the latter is a local view on gradient descent steps. Discussion on non-convex optimization is another interesting research area, but it **does not fit the circumstances and somewhat out of the scope of our work**. Moreover, the empirical observation on drifting functional directions is our motivation, **not the major contribution** of our work in LLM continual learning. We demonstrated that our method solves this observed issue and successfully mitigates catastrophic forgetting in LLM continual learning through extensive experiments.
>
> ---
>
> **Q2**: Since the paper’s core claim is that dynamically updated regularization mitigates forgetting, the experiments should explicitly probe (1) task-order sensitivity (change task orders) and (2) sequence-length robustness (Vary the number of tasks ). These ablations would substantiate that the proposed dynamic regularization is not brittle to ordering effects and remains effective over long horizons.
>
> **A2**: Thank you for your comment. However, please kindly note that **we have already conducted** these experiments you mentioned about task-order sensitivity and sequence-length robustness in Section 4 in our initial version, with continual learning performances on different task orders and sequence lengths presented in Table 1 - 4. Please refer to our paper for these results.
>
> ---
>
> Given these clarifications, we respectfully request a re-evaluation of our work.

---

> > ### Author Response · Authors · 2025-11-27
> >
> > Dear Reviewer K2j5,
> >
> > It is our great honor to have you as a reviewer for our paper. We sincerely appreciate the time and effort you have devoted to your review. We have carefully clarified or addressed all the concerns you raised, and sincerely hope that these clarifications and revisions have resolved them.
> >
> > As the discussion phase is nearing its end, we just wanted to gently check whether you have any remaining concerns. We would be very happy to clarify anything further if needed.
> >
> > Thank you again for your thoughtful feedback. Wishing you a lovely day!
> >
> > Warmly,
> >
> > Authors

---

### Official Review · Reviewer_Lsnj · 2025-11-01

**Soundness:** 3
**Presentation:** 2
**Contribution:** 1
**Rating:** 2
**Confidence:** 3

**Summary:**

This paper aims to improve a currently existing technique (LoRA) that is used for fine tuning LLMs. The existing technique forces the updates to the weights to be low rank, the proposed method ensures that the updates for new tasks are also orthogonal to previous updates.

**Strengths:**

I think the main idea of the paper is very natural: the weights are in a high dimensional space and the updates are low rank, so there is quite bit of space to force the updates to be orthogonal to each other.

**Weaknesses:**

My main concern is that paper's presentation of main idea is unnecessarily complicated. For example, it uses online PCA even when it is not really using the principal components, from equation 13, it seems that a simple gram-schmidt process would do. In other words, maintaining any orthonormal basis for the gradient updates would suffice, it doesn't have to be principal components.

The writing could be improved. For example, I find the  introduction of LoRA in section 2.2 quite abrupt and more motivation/context there would be helpful.
It would be helpful in equation 4 to first introduce the gradient notation. The references need to be fixed as well, many of them are currently just urls.

**Questions:**

Is PCA actually necessary here?
It seems to me that these techniques would also apply to general neural networks, is there a reason that the focus is on LLMs?

In equation 10, to force the sum of the two terms to be orthogonal to $\tilde{v}^k$, you set each to be orthogonal. That would be sufficient, but not necessary, could you justify it more?

---

> ### Author Response · Authors · 2025-11-21
> **Response to Reviewer Lsnj**
>
> Dear Reviewer Lsnj:
> Thank you for recognizing our work in conducting orthogonal gradient updates in a dynamic functional direction environment. But we have to note that the key to our method is **mitigating the drift of functional directions, which prior works overlook, rather than simply conducting an orthogonal LoRA update**. We apologize for potential misunderstandings. In response to your concerns, we have carefully revised the manuscript and provide detailed responses below to help you grasp the gist of our method.
>
> ---
>
> **W1&Q1**: My main concern is that paper's presentation of main idea is unnecessarily complicated. For example, it uses online PCA even when it is not really using the principal components, from equation 13, it seems that a simple gram-schmidt process would do. In other words, maintaining any orthonormal basis for the gradient updates would suffice, it doesn't have to be principal components.
>
> Is PCA actually necessary here? It seems to me that these techniques would also apply to general neural networks, is there a reason that the focus is on LLMs?
>
> **A1**: We apologize for any potential misunderstanding. We respectfully argue that our method is **not simply** an orthogonal gradient continual fine-tuning approach. We briefly recap the basic logic of our paper as follows: in Section 3.1, we identify the key issue of functional direction drift in **prior regularization-based methods**, including the orthogonal approaches. Then we mitigate this tracking issue by proposing a novel functional direction tracking method realized with online PCA in Section 3.2, on which basis we conduct orthogonal cutting in Section 3.3. Finally, we reach an optimal continual learning result in extensive experiments in Section 4.
>
> Here, we consider you may have misunderstood that online PCA is unnecessary in our method. However, as described above, online PCA is indispensable in tracking the drift of functional directions, which is crucial for the effectiveness of orthogonal cutting.
>
> ---
>
> **W2**: The writing could be improved. For example, I find the introduction of LoRA in section 2.2 quite abrupt and more motivation/context there would be helpful. It would be helpful in equation 4 to first introduce the gradient notation. The references need to be fixed as well, many of them are currently just urls.
>
> **A2**: Thank you for these suggestions. Our method is developed for continual learning using LoRA fine-tuning, so we first introduce LoRA as a preliminary. We additionally explained this in Section 2.2. As for the d notation in Equation 4, it is explained right after the equation for fluency. Also, the references are updated and are complete now.
>
> ---
>
> **Q2**: In equation 10, to force the sum of the two terms to be orthogonal to $\tilde{v}^k$, you set each to be orthogonal. That would be sufficient, but not necessary, could you justify it more?
>
> **A3**: Thank you for this valuable question. As shown in Section 3.3, we only cut the gradient to make the former one, *i.e.* $(dB)Ax$, orthogonal to $\tilde{v}^k$. As the gradient update is a continuous flow, the latter one, *i.e.,* B(dA)x, still maintains orthogonality as momentum.
>
> ---
>
> Given these clarifications, we respectfully request a re-evaluation of our work.

---

> > ### Author Response · Authors · 2025-11-27
> >
> > Dear Reviewer Lsnj,
> >
> > It is our great honor to have you as a reviewer for our paper. We sincerely appreciate the time and effort you have devoted to your review. We have carefully clarified or addressed all the concerns you raised, and sincerely hope that these clarifications and revisions have resolved them.
> >
> > As the discussion phase is nearing its end, we just wanted to gently check whether you have any remaining concerns. We would be very happy to clarify anything further if needed.
> >
> > Thank you again for your thoughtful feedback. Wishing you a lovely day!
> >
> > Warmly,
> >
> > Authors

---

### Author Response · Authors · 2025-11-23
**Summary of Revision and Invitation for Further Discussion**

We thank all reviewers again for your constructive comments and valuable suggestions. We have revised our manuscript according to your suggestions to address your concerns. Notably, we have added/extended extensive supplementary content (e.g., experiments, analyses, visualizations, or comparisons) to further address your concerns and strengthen the validity of our work. Key additions are listed below:
- **Section 2.2 (modified)**: Address the abrupt introduction of LoRA by supplementing its motivation (as the parameter-efficient foundation of our method).

- **Section 4.3 (extended)**: Add comparative experiments of Online PCA variants (IPCA, SGA, GHA, SNL) versus our modified CCIPCA, reporting tracking quality (Figure 3(a)), Average Accuracy (Table 1), BWT/FWT scores (Table 3), and computational costs (0.46s per step for CCIPCA vs. ≥0.66s for others) to verify CCIPCA’s efficiency advantage.

- **Appendix B (extended)**: Expand functional direction drift analysis from single datapoints to all CL Benchmark datasets, adding cosine similarity trends across diverse data distributions (Figure 5) to strengthen the generalization of our motivation.

- **Appendix C (modified)**: Add hyperparameter ablation experiments (tracking factor δ, residual threshold εₘₐₓ), presenting Average Accuracy results of different parameter combinations (Table 7) and detailing residual rate/coverage trends’ role in algorithm decision-making.

- **Figure 1 (modified)**: Optimize subplot annotations and explanatory text, explicitly linking each visual component to DOC’s core steps (functional direction tracking, orthogonal projection) to improve readability.

These revisions and supplements adequately address your concerns and are supported by concrete experiments or explanations. We welcome further questions or insights to refine our work and look forward to your continued guidance. Thank you again for your expertise.

---

### Author Response · Authors · 2025-11-30
**Final Remarks for Area Chair (Part 1/2)**

Dear Area Chair,

We sincerely appreciate your stepping in to oversee the review process for our submission under these unusual circumstances. We understand the significant additional workload this reassignment entails and are grateful for your time and effort in evaluating our work, the original reviews, and our rebuttal updates.

---

**Summary of Strengths**

The reviewers have highlighted the following key strengths of our work:

- **Strong & Grounded Motivation**: The method identifies the key issue that prior works overlook, *i.e.*, the drift of functional directions in the process of continual fine-tuning, which is grounded in diverse continual learning scenarios (Reviewers  K2j5, rKEH,  3uqt).
- **Novel and Intuitive Idea**: To mitigate the drift of functional directions, we propose the Dynamic Orthogonal Continual (**DOC**) fine-tuning method that tracks the drift of functional directions and conducts *effective* orthogonal updates (Reviewers  K2j5, rKEH,  3uqt).
- **Strong Empirical Results:** Extensive experiments demonstrate the effectiveness of our method in tracking functional directions and mitigating catastrophic forgetting (Reviewers rKEH,  3uqt).
- **Comprehensive Evaluation**: Experiments are extensive, covering three models and reporting the continual learning performance on diverse matrics, benchmarks, and task lengths, and include ablation studies to highlight the impact of the method. (Reviewer 3uqt).

---

**Response to Key Concerns**

1. **Substitute technical implementation**
- **Concern**: The method conducts Online PCA with the CCIPCA technique, but does not evaluate other Online PCA techniques. (Reviewer 3uqt)
- **Resolution:** We addressed this by conducting **new experiments** during the rebuttal phase to further discuss the effect of using different Online PCA techniques in our method, including IPCA, SGA, GHA, and SNL techniques. We report their tracking qualities, average Accuracies, BWT, FWT scores, and computational costs.
- **Location:** (**New Revision)** Section 4.3, Figure 3 (a), Table 1, Table 3, Table 6.

---

2. **Conditions of Motivation**
- **Concern**: The drift analyses in Figure 2 should be conducted under more diverse conditions. (Reviewer 3uqt)
- **Resolution**: We present drift analyses under all the CL Benchmark datasets to further demonstrate the pervasiveness of functional direction drift.
- **Location**: (New Revision)  Figure 5 in Appendix B.

---

3. **More varied ablation studies**
- **Concern**: Reviewers request further evaluation on how varying the tracking strength affects tracking performances and continual learning results. (Reviewer 3uqt)
- **Resolution**: We introduce DOC-strong and DOC-weak by changing the tracking strength in our method, and present their tracking performances and continual learning results.
- **Location**: (New Revision) Section 4.3, Figure 3 (b), Table 1, Table 3

4. **More combinations of hyperparameters**
- **Concern**: Our method introduces several hyperparameters in tracking the drift of the functional directions. Reviewers request studies on different combinations of them.
- **Resolution**: We present studies on different combinations of the tracking factor and the residual threshold, and demonstrate their tracking performances and continual learning results.
- **Location**:  (New Revision) Section 4.3 ablation study, and Appendix C.

5. **The necessity of tracking**
- **Concern**: Reviewers doubt the necessity of tracking functional directions. (Reviewer  **Lsnj**)
- **Resolution**: We **clarify** that it is tracking the drifting functional directions that make our orthogonal fine-tuning effective. For one thing, we identify this drift as the key issue that prior works overlook, which makes tracking necessary; for another, our ablation studies show the indispensable role of tracking in our method.
- **Location**: (Original Content) Section 3.1, 4.3.

6. **Motivation validity**
- **Concern**: Reviewers believe that the key issue in continual learning is the shifting basin in the loss landscape, not the drift of functional directions. (Reviewer K2j5)
- **Resolution**: We acknowledge that the shifting basin is another interesting research area, but it is a global view on the loss landscape, and **does not fit the circumstances of our work. Moreover, we clarify that** the issue of drifting functional directions is strongly grounded in adequate experiments in our original paper.
- **Location**: (Original Content) Section 3.1.

---

---

> ### Author Response · Authors · 2025-11-30
> **Final Remarks for Area Chair (Part 2/2)**
>
> 7. **Other Questions about technical details**
>
> The reviewers also asked minor questions about technical details. For your convenience, we've copied brief answers below:
>
> - **Q**: Whether the LoRA is shared across CL tasks (Reviewer rKEH)
>
>     **A**: The LoRA is shared across CL tasks, and we do not introduce new LoRA modules for new tasks in our method.
>
>     **Location**: (Original Content) Section 3, Algorithm 1
>
> - **Q**: The dimension issue of cosine similarity and initialization of PCA in Figure 2 (Reviewer rKEH)
>
>     **A**:  We avoid the dimension issue in our experiment in Figure 2, where we only take the principal components initialized beforehand when computing coordinate vectors.
>
>     **Location**: (Original Content) Section 3.1, Figure 2
>
> - **Q**: Applicability to other parameter-efficient-tuning CL methods (Reviewer rKEH)
>
>     **A**: We **clarify** that our method is specifically developed for LoRA continual fine-tuning, but extension to other parameter-efficient tuning can be a valuable future direction.
>
>     **Location**: (New revision) Section 2.2
>
> - **Q**: Sufficient but not necessary orthogonal cutting (Reviewer **Lsnj**)
>
>     **A**: We only cut the gradient to make part of the update orthogonal, and as the gradient update is a continuous flow, the other part still maintains orthogonality as momentum.
>
>     **Location**: (Original Content) Section 3.3

---

### Meta-Review · Area_Chair_VfCW · 2025-12-17

**Summary:**

This paper identifies an issue arising from functional direction drift in continual fine tuning of large language models, which may explain why existing regularization based continual learning (CL) methods can fail. To address this issue, the authors propose to track and modify functional directions via online PCA, while enforcing updates for new tasks to be orthogonal to previous updates in order to mitigate task interference. Experiments on LLaMA and T5 models demonstrate that the proposed method can reduce catastrophic forgetting and improve performance in terms of AA, BWT, and FWT on standard benchmark datasets.

**Reviewer Concerns:**

All reviewers agree that identifying functional direction drift is important and that the proposed use of PCA and orthogonal updates is intuitive and clearly motivated. However, several concerns were raised. Reviewer Lsnj’s main concern relates to the presentation and necessity of the online PCA component, specifically questioning whether PCA is essential, or whether maintaining an orthonormal basis alone would suffice. Although the authors attempted to address this point in their response, the explanation does not seem to directly answer the reviewer’s question regarding the necessity of extracting principal components. Reviewer K2j5 felt that some claims, particularly those related to direction drift, were overstated, and also suggested additional ablation studies. Based on my reading, the authors’ response adequately addressed these concerns. Reviewer rKEH raised presentation related questions, which were reasonably handled in the rebuttal. Reviewer 3uqt similarly questioned the rationale for online PCA and noted that comparisons with existing PCA based approaches were unclear. This reviewer also expressed concerns about the choice of baselines and the motivation experiments. Although the authors provided additional experiments in response, the comparisons remain limited to a narrow set of methods, while treating other strong PEFT based CL baselines, such as ProgPrompt, as oracle methods.

Overall, I believe the paper addresses an important and underexplored limitation of regularization based CL methods, and the proposed solution is intuitive and conceptually clear. However, I am not fully convinced by the experimental evaluation. The main comparison baselines, such as LwF (2017) and O LoRA (2023), are relatively weak, while more recent and stronger approaches, such as ProgPrompt (2023), which has quite a few strong follow up works with even better performance, is only considered as an oracle method. Although ProgPrompt requires task identities, several follow up methods remove this requirement while achieving improved results. I suggest the authors could do a more comprehensive searching Compared to ProgPrompt (2023), the proposed method performs significantly worse. A more comprehensive and up to date baseline comparison is therefore necessary.

**Reviewer Scores:**

Based on the rebuttal, I believe most concerns raised by Reviewer K2j5 and Reviewer rKEH have been satisfactorily addressed, and their scores may be adjusted. However, the key questions raised by Reviewer Lsnj and Reviewer 3uqt remain somewhat unresolved. Therefore, I recommend rejection at this time, while encouraging the authors to carefully incorporate all reviewers’ feedback and resubmit a revised version to a future conference for a more thorough evaluation.

---

### Decision · Program_Chairs · 2026-01-26

Reject